# β-arrestin1 and 2 exhibit distinct phosphorylation-dependent conformations when coupling to the same GPCR in living cells

Raphael S. Haider [1,9], Edda S. F. Matthees [1,9], Julia Drube [1], Mona Reichel[1], Ulrike Zabel[2], Asuka Inoue [3,4], Andy Chevigné [5], Cornelius Krasel[6], Xavier Deupi [7,8] & Carsten Hoffmann [1] ✉

β-arrestins mediate regulatory processes for over 800 different G protein-coupled receptors (GPCRs) by adopting specific conformations that result from the geometry of the GPCR–β-arrestin complex. However, whether β-arrestin1 and 2 respond differently for binding to the same GPCR is still unknown. Employing GRK knockout cells and β-arrestins lacking the finger-loop-region, we show that the two isoforms prefer to associate with the active parathyroid hormone 1 receptor (PTH1R) in different complex configurations ("hanging" and "core"). Furthermore, the utilisation of advanced NanoLuc/FlAsH-based biosensors reveals distinct conformational signatures of β-arrestin1 and 2 when bound to active PTH1R (P-R*). Moreover, we assess β-arrestin conformational changes that are induced specifically by proximal and distal C-terminal phosphorylation and in the absence of GPCR kinases (GRKs) (R*). Here, we show differences between conformational changes that are induced by P-R* or R* receptor states and further disclose the impact of site-specific GPCR phosphorylation on arrestin-coupling and function.

Arrestins were initially described as proteins that terminate primary G protein-coupled receptor (GPCR) signalling. In recent years it became evident that arrestins not only serve as a steric hindrance of further G protein activation but can interact with more than 100 different proteins[1] to modulate alternative GPCR signalling (e.g. mitogen-activated protein kinase[2–4]), internalisation and trafficking (e.g. clathrin, adaptor protein 2[5,6]).

Structural biology studies identified two major interaction interfaces between the receptor and β-arrestin, which are comprised of the arrestin N-domain binding to phosphorylated receptor domains[7–9] and the finger-loop-region (FLR) inserting into the intracellular GPCR cavity[10–13]. Additionally, recent studies showed that the C-edge loops of arrestins can act as a membrane anchor to stabilise GPCR–arrestin complexes[12,14]. As postulated before, this implies a multistep binding

[1]Institut für Molekulare Zellbiologie, CMB—Center for Molecular Biomedicine, Universitätsklinikum Jena; Friedrich-Schiller-Universität Jena, Hans-Knöll-Straße 2, D-07745 Jena, Germany. [2]Institut für Pharmakologie und Toxikologie, Universität Würzburg, Versbacherstraße 9, D-97078 Würzburg, Germany. [3]Graduate School of Pharmaceutical Sciences, Tohoku University, Sendai, Miyagi 980-8578, Japan. [4]Japan Science and Technology Agency (JST), Precursory Research for Embryonic Science and Technology (PRESTO), Kawaguchi, Saitama 332-0012, Japan. [5]Immuno-Pharmacology and Interactomics, Department of Infection and Immunity, Luxembourg Institute of Health (LIH), Esch-sur-Alzette, Luxembourg. [6]Philipps-Universität Marburg; Fachbereich Pharmazie; Institut für Pharmakologie und Klinische Pharmazie, Karl-von-Frisch-Str. 1, 35043 Marburg, Germany. [7]Laboratory of Biomolecular Research, Paul Scherrer Institute, CH-5232 Villigen, Switzerland. [8]Condensed Matter Theory Group, Paul Scherrer Institute, CH-5232 Villigen, Switzerland. [9]These authors contributed equally: Raphael S. Haider, Edda S. F. Matthees. ✉e-mail: carsten.hoffmann@med.uni-jena.de

mechanism of arrestins[15], which requires all three interaction sites for the formation of a high-affinity complex.

The structural elucidation of a growing number of GPCR–β-arrestin complexes[10,12,13,16,17] revealed two different modes of arrestin binding. In particular, the association of β-arrestin with the phosphorylated GPCR C-terminus only ('hanging' complex[8,16]) and a tight complex configuration employing additional interactions via the FLR and C-edge loops (membrane-anchored 'core' complex[10,12,13,17]) can be distinguished. Both complex configurations were shown to be functionally active with one decisive difference: only the GPCR–β-arrestin complex in the 'hanging' configuration allows further G protein activation[8,18], while the 'core' complex obstructs the G protein binding interface.

Using nuclear magnetic resonance or β-arrestin2 biosensors, arrestins have been shown to adopt distinct active conformations in order to match the individual structure and phosphorylation state of the bound GPCR[10,19–23]. Depending on the nature of the resulting β-arrestin conformation, a certain set of effector proteins might then be recruited to the complex. Via this process, the two β-arrestin isoforms are able to control targeted functions for a plethora of different GPCRs. However, β-arrestin1 and 2 share high sequence and structural similarities and little is known about their differential regulation by a specific GPCR.

In this study, we comprehensively investigate the activation of β-arrestin1 and 2 via the parathyroid hormone 1 receptor (PTH1R). We chose this class B model GPCR due to its interesting features regarding intracellular trafficking and signalling from endosomes via cAMP[24–27], as well as the robust recruitment of both β-arrestin isoforms[28] (see also ref. [29]).

Recently, the PTH1R has been shown to elicit an anti-inflammatory effect in testis, which makes the receptor an interesting target for pharmacological intervention in orchitis[30]. Intriguingly, it was shown that this effect is selectively mediated by Gq and β-arrestin1, while β-arrestin2 did not influence this particular signalling in vivo. This finding also raises the question whether the two β-arrestin isoforms adopt distinct active conformations upon binding to the same GPCR, in order to trigger specific signalling responses.

In this study, we present generally improved intramolecular BRET biosensors for β-arrestin2, based on our previous work[19], and a complete design of biosensors for β-arrestin1, which enable us to assess the conformational changes of both isoforms in living cells. We apply these sensors to elucidate differences in conformational change between β-arrestin1 and 2, induced by association with the same GPCR. Furthermore, we investigate whether differential receptor phosphorylation patterns are responsible for the formation of specific functional β-arrestin conformations. This analysis additionally includes the utilisation of GRK2, 3, 5 and 6 quadruple knockout cells (ΔQ-GRK[29]), allowing us to assess the differential impact of the active and phosphorylated receptor (P-R*) or just the active receptor (R*), independent of GRK-mediated phosphorylation, on β-arrestin1 and 2 conformations.

## Results

### The configuration of a GPCR–β-arrestin complex determines its functionality

β-arrestin conformational changes are directly induced by association with the membrane receptor. Hence, the differently engaged binding interfaces and the longevity of the resulting complex finally define the active state of arrestin. Thus, it was pivotal for us to first evaluate the binding modes between the PTH1R and β-arrestin1 and 2. Here, we focussed on three distinct complex configurations: the canonical 'core' complex, the 'hanging' complex and a complex that forms independently of GRK phosphorylation (Fig. 1a).

To assess the formation of these different complex configurations (Fig. 1a) and examine whether β-arrestin1 and 2 preferably utilise different binding interfaces, we performed intermolecular NanoBRET recruitment assays in HEK293 cells (HEK-WT) or GRK2, 3, 5 and 6 quadruple knockout cells (ΔQ-GRK[29]) for wild type (WT) β-arrestins and for constructs lacking the FLR (dFLR). The concentration-dependent association of arrestins to the PTH1R in these conditions upon application of parathyroid hormone (1-34) (PTH(1-34)) is shown in Fig. 1b.

The recruitment measured with β-arrestin WT constructs in HEK-WT cells is indicative of 'core' complexes that are formed with the phosphorylated and active PTH1R (P-R*, Fig. 1a), as all interaction interfaces are accessible. Since the β-arrestin-dFLR mutants lack essential amino acids for the binding of the intracellular receptor cavity (β-arrestin1: Y63 to K77, β-arrestin2: Y64 to K78), we interpret recruitment of those constructs as the capability to form a 'hanging' GPCR–β-arrestin complex[8]. In the 'hanging' complex configuration β-arrestins mostly interact with negatively charged or phosphorylated residues of the receptor C-terminus[16], thus we can interpret the measured binding as a complex formation with the P-R receptor state (Fig. 1a), even though the ligand is present. In contrast, the recruitment measured in ΔQ-GRK cells reflects the affinity of the respective β-arrestin toward the receptor, independent of GRK phosphorylation (R*, Fig. 1a).

Both β-arrestin isoforms showed a robust recruitment toward the P-R* receptor state (Fig. 1b). Importantly, deletion of the respective FLRs and GRKs from our cellular system drastically reduced measured BRET changes for the recruitment of both β-arrestin isoforms. Yet we found detailed differences between β-arrestin1 and 2, as the deletion of the FLR differentially influenced their recruitment (Fig. 1b the concentration-response curves are depicted again in Supplementary Fig. 1a–c with individual axis limits). The β-arrestin2-dFLR mutant produced only a fraction of the signal (~7 % of β-arrestin2 WT) compared to the β-arrestin1-dFLR (~36 % of β-arrestin1 WT), leading us to the conclusion that β-arrestin1 is better suited to form a 'hanging' complex in comparison to β-arrestin2 (Fig. 1b and Supplementary Fig. 1b). Interestingly, the β-arrestin2 recruitment measured in absence of ubiquitously expressed GRKs, using ΔQ-GRK cells, shows similarly reduced values as the signal obtained from the β-arrestin2-dFLR measurements (Fig. 1c). This implies that both GRK-mediated receptor phosphorylation, as well as the FLR interaction interface play a role for high-affinity β-arrestin2 association with the PTH1R. In contrast, the β-arrestin1 recruitment in ΔQ-GRK cells shows reduced values in comparison to the β-arrestin1-dFLR recruitment (Fig. 1c, $p = 0.068$), suggesting that β-arrestin1 relies more prominently on receptor phosphorylation to form stable complexes with the PTH1R.

Here, it is important to note that the analysed binding interfaces cannot necessarily be considered as separate entities, since both of them constitute essential parts of GPCR–β-arrestin complexes. We hypothesise that all binding interfaces contribute to high-affinity β-arrestin1 and 2 recruitment and specifically, since the conformational state of the FLR has been shown to be influenced by proximal receptor phosphorylation[31–33], our measurement system might not be able to differentiate combined effects. Nevertheless, the presented results indicate that β-arrestin1 and 2 seem to utilise the analysed binding interfaces differently. Hence, these findings provide evidence that β-arrestin1 and 2 form similar, yet distinct complexes when coupling to the PTH1R, as β-arrestin1 seems to prefer to engage the phosphorylated GPCR C-terminus and β-arrestin2 still shows considerable recruitment to the GPCR independently of GRK phosphorylation (Supplementary Fig. 1a–c). This has already been proposed in structural and computational studies[34,35]. An interesting structural explanation for this general phenomenon could be the rather flexible β-strand XIV inside the β-arrestin2 C-domain, which is shorter than in β-arrestin1[35]. This structural feature might help β-arrestin2 to accommodate the energetic preconditions in order to break stabilising polar

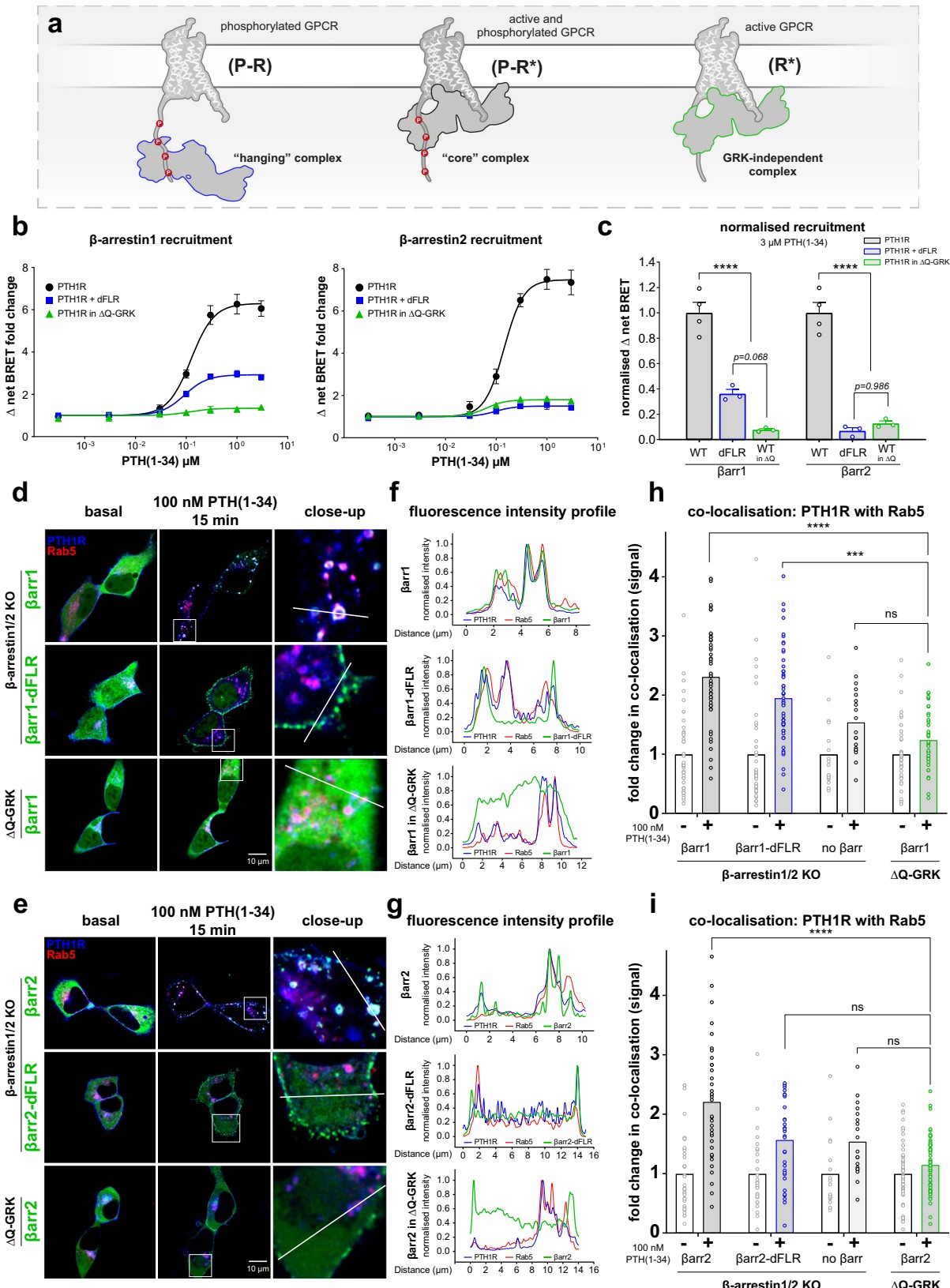

core interactions and engage the open GPCR cavity with a lower number of charged residues at the GPCR C-terminus.

Having established the differently engaged complex configurations between the PTH1R and the two β-arrestin isoforms, we wanted to assess their cellular localisation and get a first measure of their functionality. Thus, we employed confocal life-cell microscopy, using

β-arrestin1/2 knockout[36] or ΔQ-GRK cells[29]. Cells were transfected with PTH1R-CFP, β-arrestin-YFP and the early endosome marker Rab5-mCherry and stimulated with 100 nM PTH(1-34) for 15 min. Both β-arrestin isoforms, as well as the dFLR variants showed translocation upon agonist stimulation of the receptor (Fig. 1d, e, Supplementary Fig. 1 and Supplementary movies 1–6).

**Fig. 1 | The configuration of specific PTH1R–β-arrestin complexes determines their functionality. a** Schematic depiction of the 'hanging', 'core' and GRK-independent GPCR-β-arrestin complex configurations. **b** NanoBRET-measured recruitment of β-arrestin1 and 2 to the PTH1R upon stimulation with indicated concentrations of PTH(1-34). Curves show the recruitment measured for β-arrestin WT constructs in HEK293 cells (HEK-WT), analogous experiments performed in ΔQ-GRK cells and recruitment measurements of β-arrestin-dFLR mutants in HEK-WT. Results are shown as Δ net BRET fold change, mean of at least three independent repetitions (βarr-dFLR and βarr in ΔQ-GRK $n = 3$; βarr-WT $n = 4$) ± SEM. **c** Data correspond to the BRET changes at saturating ligand concentration from **b**, normalised to the respective β-arrestin WT condition. To test for significance between β-arrestin association in a 'hanging' or GRK-independent complex, a one-way ANOVA, followed by a two-sided Tukey's test (*, $p < 0.05$; **, $p < 0.01$; ***, $p < 0.001$; ****, $p < 0.0001$). For both β-arrestin 1 and 2, WT vs respective βarr-dFLR and βarr in ΔQ-GRK $p < 0.0001$. **d** and **e** Representative live-cell confocal microscopy images of β-arrestin1/2 double knockout and ΔQ-GRK cells transfected with PTH1R-CFP (blue), the early endosome marker Rab5-mCherry (red) and the respective β-arrestin-YFP WT or dFLR constructs (green). Additionally, close-ups of the stimulated conditions are shown, which correspond to the white squares indicated in the representative image. Images were acquired before and after stimulation with 100 nM PTH(1-34) for 15 min from at least three cover slips, prepared from at least three independent transfections ($n \geq 3$). **f** and **g** normalised fluorescence intensity profiles of all three acquired channels along the white line indicated in the close-up images in d or e, respectively. **h** and **i** The quantification of co-localisation of PTH1R-CFP with Rab5-mCherry in β-arrestin1/2 knockout and ΔQ-GRK cells was calculated using Squassh and SquasshAnalyst (number of images per respective condition; βarr1 (39), βarr1-dFLR (43), no βarr (17), βarr1 in ΔQ-GRK (38), βarr2 (33), βarr2-dFLR (27), βarr2 in ΔQ-GRK (50)) and is represented as mean fold change in co-localisation signal + SEM. Statistical significance was calculated by two-way ANOVA, followed by a post-hoc comparison with Bonferroni correction (*, $p < 0.05$; **, $p < 0.01$; ***, $p < 0.001$; ****, $p < 0.0001$).Complete results of the statistical analysis are shown in Supplementary Table 1. Source data are provided as a source data file.

WT β-arrestin1 and 2 exhibit stable co-localisation with the receptor (quantification shown in Supplementary Fig. 1d–g) and can be found in intracellular compartments alongside Rab5 (Fig. 1d, e). In contrast, the β-arrestin-dFLR mutants translocate to the membrane (Fig. 1d, e) but show reduced co-internalisation (Supplementary Fig. 1d–g). To support these findings, close-up representations of the stimulated images (magnified sections indicated by white squares) are shown in the third columns of Fig. 1d, e, respectively. Additionally, the fluorescence intensity profiles of all three acquired channels along the indicated white lines in the close-up images are plotted in Fig. 1f, g. Especially the fluorescence intensity profiles for β-arrestin1 and 2 in ΔQ-GRK show that PTH(1-34) stimulation still induces a slight membrane recruitment of β-arrestin2, while there is no observable translocation of β-arrestin1. These findings indicate that the translocation of β-arrestin1 is strictly dependent on GRK-mediated receptor phosphorylation, in contrast to β-arrestin2. Furthermore, based on these observations, it is tempting to speculate that 'hanging' and 'core' complexes may preferentially reside in different membranous compartments.

Next, we assessed the co-localisation between the PTH1R and Rab5, as a surrogate measurement for receptor internalisation and early trafficking. Analysis of this value in the presence or absence of arrestins and GRKs would allow us to evaluate the function of these proteins in PTH1R internalisation. Here, the PTH(1-34)-induced co-localisation between the receptor and the early endosome marker was significantly decreased in the absence of β-arrestins (Fig. 1h, i, data for 'no β-arr' condition shown twice to enable direct comparison, the comprehensive results of statistical testing for these data can be accessed in Supplementary Table 1) when compared to re-expression of either β-arrestin1 or 2 in β-arrestin1/2 knockout cells. Strikingly, overexpression of the β-arrestin1-dFLR mutant in β-arrestin1/2 knockout cells significantly increased the co-localisation between the PTH1R and Rab5 in comparison to the drastically reduced values recorded for β-arrestin1 overexpression in ΔQ-GRK cells (Fig. 1h, $p = 0.0002$), whereas the β-arrestin2-dFLR variant was unable to further support receptor internalisation (Fig. 1i, $p = 0.0887$). Intriguingly, the use of ΔQ-GRK cells yielded no significant internalisation (Fig. 1h, i and Supplementary Table 1). These measurements confirm that β-arrestins and GKRs have to act synergistically on activated GPCRs to mediate receptor internalisation. Additionally, our results imply that 'hanging' complexes of β-arrestin1 and 2 differ in their functionality to facilitate the endosomal localisation of bound PTH1R. To be able to draw these conclusions, we additionally analysed specific segmentation parameters for the three fluorophore-labelled proteins of interest. Hence, Supplementary Fig. 1h, i shows the similar mean object intensities and integrated signal per cell size for PTH1R-CFP, β-arrestin-YFP and Rab5-mCherry between the basal images of all tested conditions.

With this, our confocal microscopy analysis demonstrates an additional difference between the two β-arrestin isoforms, as we conclude that the 'hanging' complex between β-arrestin1 and the PTH1R is still functional with respect to internalisation, in contrast to β-arrestin2. This is in line with the findings of ref. 9. However, our analysis allowed us to compare the functionality of the β-arrestin2-dFLR mutant with its β-arrestin1 counterpart directly.

### β-arrestin1 and 2 display different conformational change signatures upon recruitment to the same GPCR

Previous studies provide evidence that β-arrestin2 adopts different conformations upon binding to specific GPCRs or phosphopeptides[10,19–21]. To our knowledge, whether the same GPCR would induce different conformational changes for β-arrestin1 and 2 was not assessed to this point. To address this question, we now present a comprehensive conformational change biosensor design for both β-arrestin isoforms (Fig. 2).

Based on our previously published work on β-arrestin2[19], we now constructed NanoLuc luciferase (NanoLuc)- and fluoresceine arsenical hairpin binder (FlAsH)-based biosensors for both β-arrestin1 and 2 (Fig. 2a). Notably, a β-arrestin2 biosensor design using the same BRET donor and acceptor combination, but with a different configuration was recently described[37]. The recruitment of all used β-arrestin conformational change biosensors to the PTH1R was assessed in order to confirm their functionality (Supplementary Fig. 2). Since FlAsH constructs at the positions F6 and F8 showed no interaction with the receptor, they were omitted from this study.

All conformational change measurements in this study were performed in a concentration-dependent manner and can be accessed in Supplementary Figs. 3 and 4. To further characterise the developed biosensors, we recorded the emission spectra of representative F5 sensors for both β-arrestin isoforms in living cells (Fig. 2b), using the same transfection scheme as for the conducted conformational change measurements, thus including the co-transfection of the PTH1R. Without FlAsH labelling, the spectra show virtually identical, monodisperse emission peaks that correspond to the luminescence of the NanoLuc energy donor (blue). Via the addition of the acceptor fluorophore by FlAsH labelling of the analysed cells, an efficient energy transfer can be observed (green), which is reduced upon ligand addition (red). Moreover, we analysed the time-dependent conformational change (Supplementary Fig. 5a, b) and Z-factors[38] (Supplementary Fig. 5c, d) of the F5 sensors for both β-arrestin isoforms and found that both sensors feature a Z-factor > 0.5, which indicates that they are suitable for high throughput screening[38].

As an example, the concentration-dependent recruitment and conformational changes of the β-arrestin-F1, -F2 and -F3 constructs are shown in Fig. 2c. These data show that the utilised biosensors are

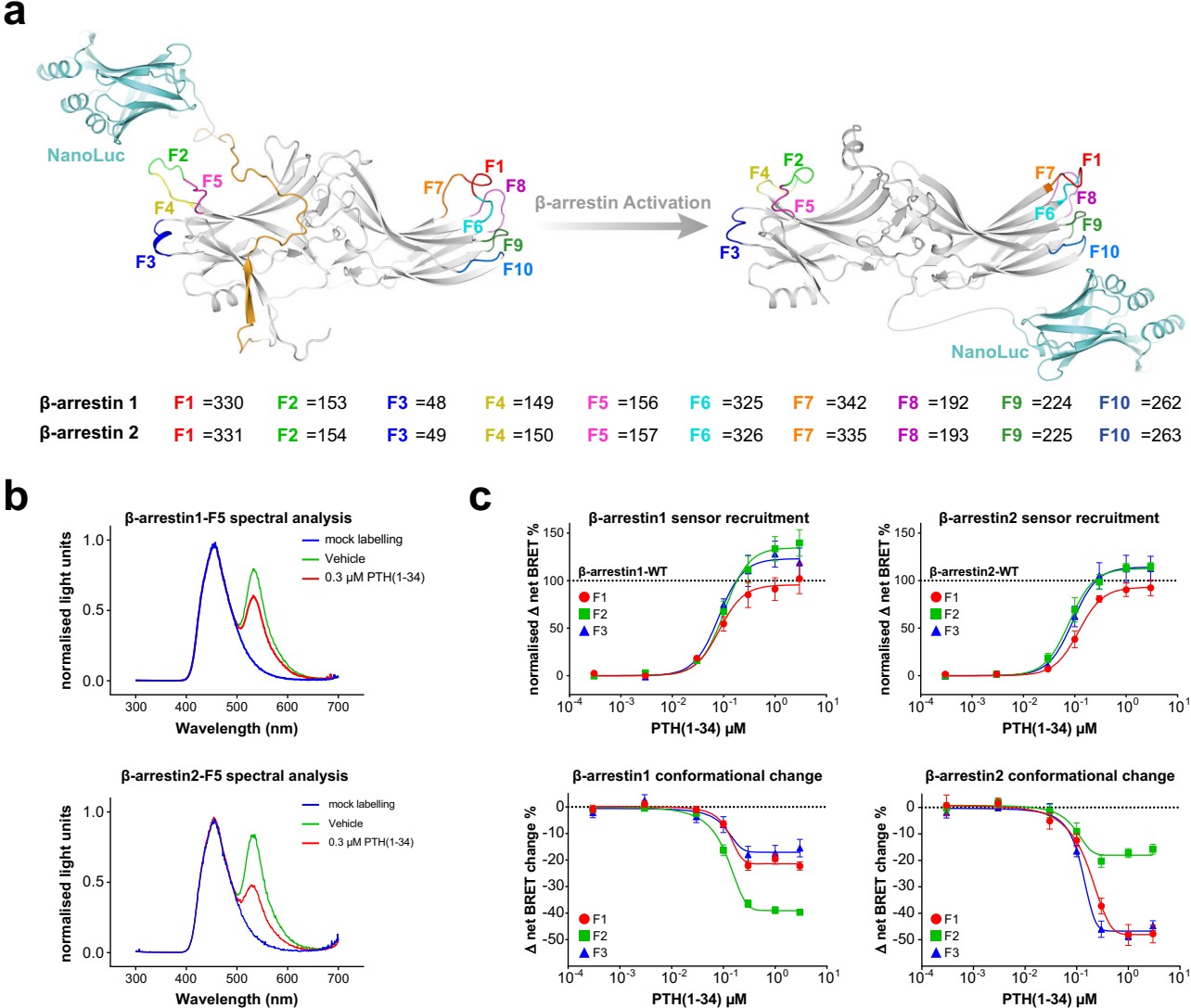

**Fig. 2 | Generation and functionality of β-arrestin1 and 2 NanoBRET conformational change biosensors. a** Overall sensor design of intramolecular β-arrestin1 and 2 conformational change biosensors. The NanoLuc BRET donor is genetically fused to the arrestin C-terminus and individual FlAsH-binding motifs are introduced at ten different positions in outward-facing loops of the arrestin N- and C-domains (F1 – F10, coloured loops and amino acid sequence positions denoted for β-arrestin-1 and 2). In the inactive state (PDB: 1CF1, with a modelled C-terminus depicted in ochre), the NanoLuc is in close proximity of the individual FlAsH-binding motifs, resulting in an efficient energy transfer. Upon GPCR- and subsequent arrestin activation (PDB: 5W0P), the distance between the BRET pair changes, depending on the nature of the conformational change and the specific labelling position. **b** Characterisation of β-arrestin1/2-F5 conformational change biosensors. Shown are the emission spectra of the F5 conformational change biosensors for β-arrestin1 and 2. The spectra were measured with and without FlAsH labelling (mock, blue), in the basal state (vehicle, green) and after ligand addition

(0.3 μM PTH(1-34), red) of the co-transfected PTH1R. **c** Concentration-dependent recruitment and conformational change of the β-arrestin1 and 2 F1, F2 or F3 biosensors upon activation of the PTH1R-WT are shown as examples. Recruitment of the individual conformational change biosensors was assessed by co-transfection of a PTH1R-HaloTag expression construct and the measurement of intermolecular BRET upon stimulation with PTH(1-34). Recruitment data are depicted as Δ net BRET %, normalised to the maximum recruitment of the respective β-arrestin WT construct. For the generation of β-arrestin conformational change data, HEK-WTcells were transfected with an untagged PTH1R-WT expression construct and one β-arrestin conformational change biosensor, FlAsH-labelled and stimulated with PTH(1-34). Conformational change data are shown as Δ net BRET change in per cent. All results are shown as mean of three independent repetitions ($n = 3$) ± SEM. All recorded recruitment and conformational change data can be comprehensively assessed in Supplementary Figs. 2, 3 and 4. Source data are provided as a source data file.

recruited to the receptor in a WT-like fashion while yielding distinct BRET changes for conformational changes, depending on the positioning of the introduced FlAsH binding site. Moreover, the measured F1, F2 and F3 conformational change data show major differences between β-arrestin1 and 2.

Consecutively, we recorded concentration-dependent conformational change signatures for all β-arrestin1 and 2 biosensors upon PTH1R stimulation with PTH(1–34) (Supplementary Fig. 3). Here, we observed effective ligand concentrations (EC$_{50}$ values) to be similar for all biosensors of both β-arrestin isoforms

(Supplementary Tables 2 and 3). Hence, we are now able to present the complete conformational change fingerprints for β-arrestin1 and 2 upon coupling to the PTH1R (Fig. 3a–c). To simplify the comparison of conformational change signatures, the mean Δ net BRET changes at saturating ligand concentrations are depicted as bar charts divided into FlAsH sensors located in the N- (Fig. 3a) and C-domain (Fig. 3c) of the respective β-arrestin isoforms. Additionally, a radar-chart representation, as well as a surface projection of the obtained conformational change data (normalised to the respective F10 biosensor data in each dataset, as it shows similar

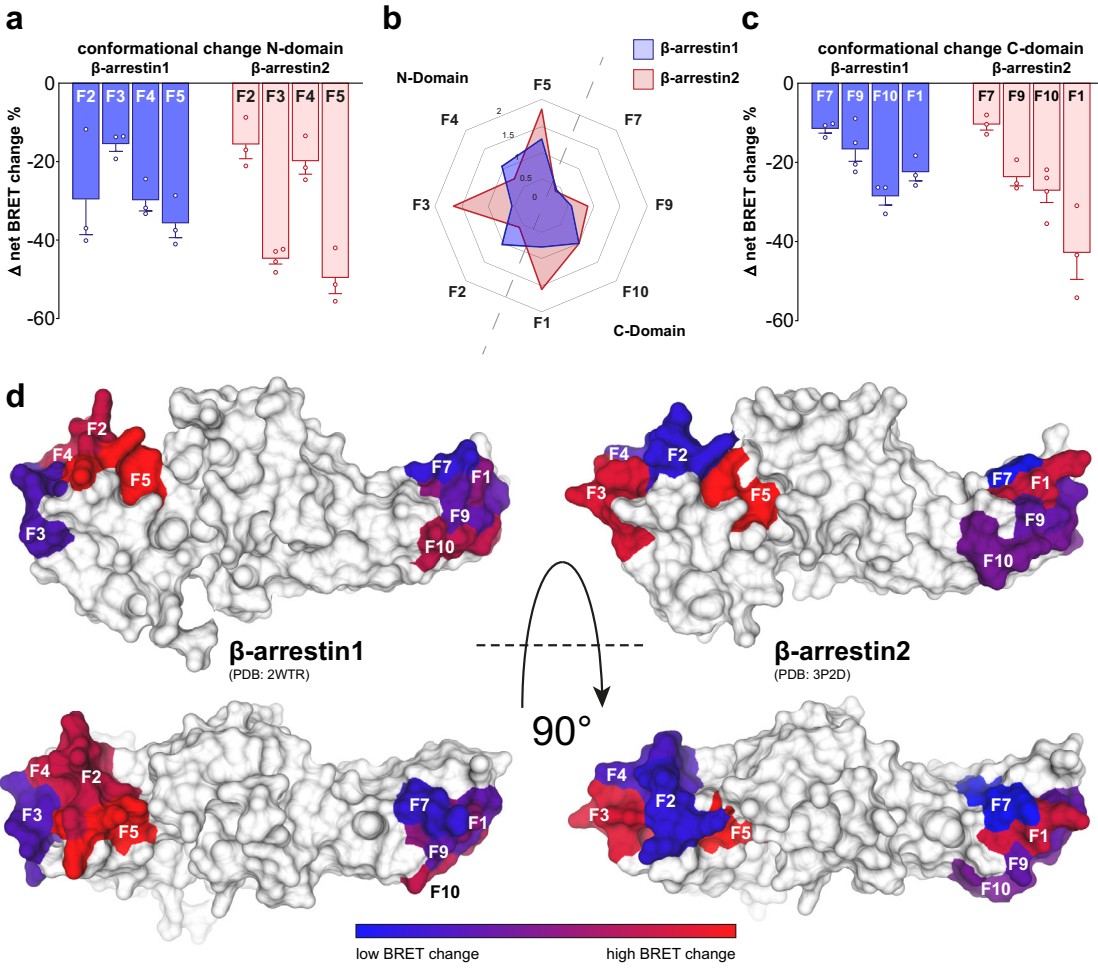

**Fig. 3 | β-arrestin1 and 2 display different conformational change signatures upon coupling to the PTH1R. a** Depiction of conformational changes induced by PTH1R activation for β-arrestin1 and 2 at 3 μM PTH(1-34) for sensors located in the respective N-domains. The β-arrestin conformational change data were generated by transfection of HEK-WT cells with an untagged PTH1R-WT expression construct and one β-arrestin conformational change biosensor, FlAsH-labelling and stimulation with 3 μM PTH(1-34). Results are shown as Δ net BRET change in percent, mean of at least three independent repetitions (β-arrestin2-F3 $n = 4$, all other sensors $n = 3$) ± SEM. **b** Radar chart representation of the data from **a** and **c**, normalised to the respective F10 biosensor values in each dataset, as this particular sensor shows

similar conformational change signals for both β-arrestin isoforms. **c** Shows conformational change measurements of β-arrestin1 and 2 sensors located in the respective C-domains, analogous to the N-domain data presented in **a**. Results are shown as Δ net BRET change in percent, mean of at least three independent repetitions (β-arrestin1-F9 and β-arrestin2-F10 $n = 4$, all other sensors $n = 3$) ± SEM. **d** Surface projection of the measured conformational change data onto crystal structures of inactive β-arrestin1 (PDB: 2WTR) and β-arrestin2 (PDB: 3P2D). The Δ net BRET change in percent is plotted on loop (-fragments), harbouring the respective FlAsH site as spectrum ranging from blue to red. Source data are provided as a source data file.

conformational change values for β-arrestin1 and 2) are shown in Fig. 3b, d, respectively.

Our data reveal major conformational differences for β-arrestin1 and 2 in their phosphate-sensing N-domains (Fig. 3a, b). Especially the F2, F3 and F4 sensor positions responded diametrically different. Here, the F2 and F4 biosensors exhibited considerably lower BRET changes for β-arrestin2, while F3 showed reduced values for β-arrestin1. This suggests that β-arrestin1 and 2 interact with the PTH1R C-terminus in distinct complexes.

Within the C-domains of β-arrestin1 and 2, we observed a higher degree of similarities for conformational changes (Fig. 3c). Signals obtained from the F7, F9 and F10 sensors, located within the outward loops of the respective C-domains, yield a similar signature for both β-arrestin isoforms. In contrast to these striking similarities, conformational changes recorded for the F1 position resulted in vastly different signal amplitudes for β-arrestin1 and 2. The FlAsH site for these constructs is located in the so-called C-edge loop 2, a supposed membrane anchor for receptor-bound arrestin[12,14]. As this loop has been shown to

play different roles for GPCR–arrestin complex configurations[10,12], our results suggest that the two β-arrestin isoforms differentially engage the GPCR to form distinct complexes. Ghosh et al. approximated differences between β-arrestins to be located within the C-domain[39], we further narrowed this finding down to the region of the F1 position (C-edge loop 2).

These findings, enabled by utilisation of these homologous conformational change biosensors, provide strong experimental evidence that β-arrestin1 and 2 indeed undergo different conformational changes when binding to the same GPCR in living cells.

### β-arrestin1 and 2 exhibit different requirements of C-terminal receptor phosphorylation for high-affinity binding

As the two β-arrestin isoforms exhibited distinct conformational changes for the interaction with the PTH1R, we hypothesised that specific receptor phosphorylation patterns might affect β-arrestin1 and 2 differentially. Hence, we investigated two phosphorylation-deficient mutants of the PTH1R. These previously published variants of

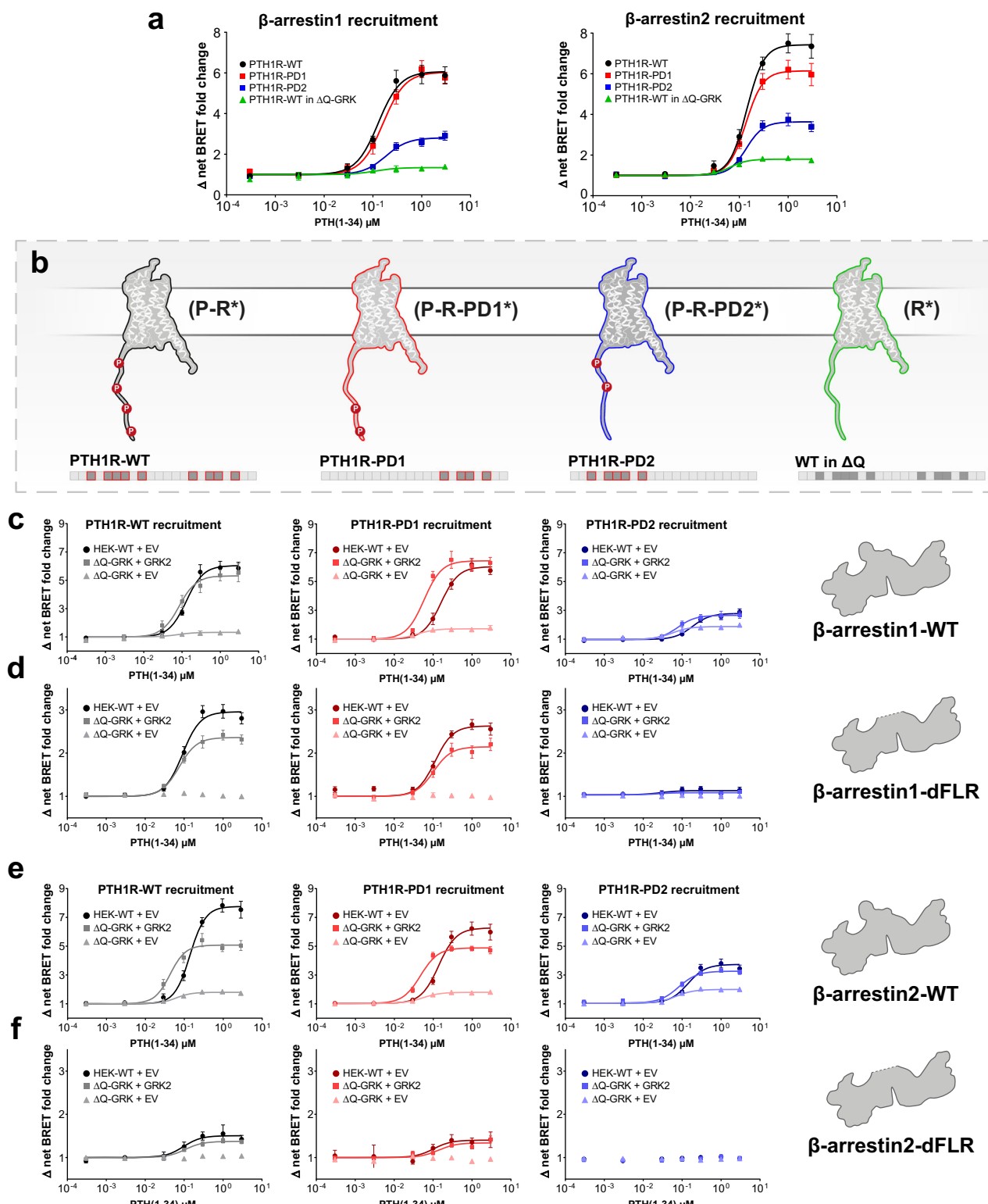

the receptor[40] were generated by alanine substitution of either a proximal (PD1) or distal (PD2) C-terminal phosphorylation cluster, respectively. Furthermore, we also examined the influence of the PTH1R in the absence of GRK-mediated phosphorylation (R*). To induce this specific receptor state, we performed the measurements in ΔQ-GRK cells using the PTH1R-WT construct. The recruitment of β-arrestin1 and 2 to these receptor phosphorylation states, as well as a schematic depiction of our complete approach to assess the impact of C-terminal GPCR phosphorylation are shown in Fig. 4a, b. Additionally,

a visualisation of β-arrestin interactions with PTH1R-WT, -PD1 and -PD2, utilising crystal structures of complexes with the V2pp (PDB: 4JQI)[7] and the CXCR7pp (PDB: 6K3F)[41] as well as a C-terminal alignment is depicted in Supplementary Fig. 6.

All three PTH1R variants recruit β-arrestin1 and 2 upon PTH(1-34) application in a concentration-dependent manner (Fig. 4a, PTH1R-WT data in HEK-WT and ΔQ-GRK from Fig. 1b are shown again to enable direct comparison). The two receptor mutants exhibited a stepwise reduction of β-arrestin2 binding, with the recruitment being more

**Fig. 4 | The PTH1R phosphorylation state differentially affects recruitment of β-arrestin1 and 2 and the resulting complex configurations. a** and **b** Overall recruitment and schematic depiction of our approach to assess phosphorylation-specific PTH1R β-arrestin activation. For the PTH1R-PD1 mutant receptor, all series of the proximal phosphorylation cluster were substituted by alanine (S498A, S500A, S501A, S502A, S504A). Similarly, phosphorylatable side chains of serines and threonines within the distal phosphorylation cluster were removed via alanine substitution (S510A, T512A, S513A, T515A) to generate PTH1R-PD2. **a** NanoBRET assessed recruitment of β-arrestin1 and 2 to the three receptor variants. Briefly, HEK-WT or ΔQ-GRK cells were transfected with either PTH1R-WT, PTH1R-PD1, or PTH1R-PD2 coupled to a C-terminal HaloTag and β-arrestin1- or 2-NanoLuc expression constructs. Upon stimulation with PTH(1-34), the concentration-dependent change in BRET signal was measured. Data are shown as Δ net BRET change in percent and represented as the mean of at least three independent repetitions (PTH1R-WT $n = 4$; all other conditions $n = 3$) ± SEM and normalised to PTH1R-WT recruitment in HEK-WT. The targeted receptor states are shown in **b**: active and phosphorylated PTH1R (P-R*), two phosphorylation-deficient receptor mutants (P-R*-PD1/PD2) or the active receptor independent of GRK phosphorylation (R*). Recruitment to PTH1R-WT, -PD1 and -PD2 in the presence (HEK-WT) and absence (ΔQ-GRK) of ubiquitously expressed GRKs, or after individual over-expression of GRK2 in ΔQ-GRK. The data are shown for β-arrestin1 (**c**) and the β-arrestin1-dFLR mutant (**d**) and analogously for β-arrestin2 (**e**) and the β-arrestin2-dFLR mutant (**f**). PTH1R-WT recruitment (Fig. 1b) was shown again to ensure comparability. A complete panel of the GRK-specific recruitment assay in ΔQ-GRK for β-arrestin1 and 2, as well as dFLR mutants, is shown in Supplementary Fig. 7. Data are shown as Δ net BRET change fold change and represented as the mean of at least three independent repetitions (PTH1R-WT in HEK-WT + EV $n = 4$, all other conditions $n = 3$) ± SEM. Source data are provided as a source data file.

prominently obstructed for PTH1R-PD2. Interestingly, β-arrestin1 recruitment was not reduced for PTH1R-PD1, yet PTH1R-PD2 showed an attenuation of recruitment analogous to β-arrestin2. This analysis reveals yet another difference between β-arrestin1 and 2 regarding their respective requirement for specific C-terminal receptor phosphorylation: both phosphorylation clusters affect high-affinity β-arrestin2 binding, whereas proximal phosphorylation only plays a minor role for β-arrestin1 interactions with the PTH1R.

To assess whether the utilised PTH1R constructs are differently targeted by GRKs, we investigated the GRK-specific β-arrestin recruitment of the PTH1R-WT, -PD1 and -PD2 variants. Moreover, we hypothesised that the presence or absence of either the proximal or distal phosphorylation cluster would have an effect on the formed β-arrestin complex configurations. Hence, we measured the GRK-specific recruitment of the β-arrestin1 and 2 WT constructs and dFLR mutants (Fig. 4c, d and Supplementary Fig. 7).

To achieve this, we recorded the β-arrestin recruitment for PTH1R-WT, -PD1 and -PD2 in absence (ΔQ-GRK) and in presence of endogenously expressed GRKs (HEK-WT) and additionally analysed the impact of single GRK2, 3, 5 or 6 overexpression in ΔQ-GRK (combined in Supplementary Fig. 7). This assay features an analogous setup as the experiments performed in ref. 29. Additionally, we quantified the amount of GRK overexpression (Supplementary Fig. 8) and found that there were no differences between β-arrestin1 and β-arrestin2 conditions.

All tested GRK isoforms are able to mediate β-arrestin1 recruitment to PTH1R-WT, -PD1 and -PD2 at least to the same extend as in HEK-WT (Supplementary Fig. 7c, ΔQ-GRK + GRK2 is shown as an example in Fig. 4c). On the other hand, the overexpression of a single GRK could not fully rescue the ΔQ-GRK phenotype for β-arrestin2 recruitment to PTH1R-WT (Fig. 4e, Supplementary Fig. 7e), although the GRK expression levels were indistinguishable for β-arrestin1 and 2. Here, we observed a slight left shift of the concentration-dependent β-arrestin1 and 2 WT recruitment for conditions that feature kinase overexpression, when compared to the data that was recorded in HEK-WT cells (Supplementary Fig. 7c, e). While it is reasonable to expect elevated GRK levels to facilitate a robust GPCR–β-arrestin interaction at lower agonist concentrations, it is quite intriguing that the β-arrestin-dFLR mutants do not exhibit this behaviour. These experiments also clarified that the PTH1R-WT, as well as the two phosphorylation-deficient variants of the receptor show identical β-arrestin recruitment when measured in ΔQ-GRK without the overexpression of GRKs (Fig. 4c, e). Thus, we can conclude that GRKs are the main kinases to facilitate PTH1R phosphorylation and propose that other intracellular kinases may only play a secondary role, regarding the high-affinity binding of β-arrestins to PTH1R.

For GRK6 overexpression in ΔQ-GRK we noticed an increased recruitment of β-arrestins for PTH1R-PD2 compared to the other GRK subtypes (Supplementary Fig. 7). It is tempting to speculate that the proximal phosphorylation cluster exhibits a preference for GRK6.

However, our current cluster approach does not allow an individual assignment to phosphorylation sites, especially since we have no information of $k_{cat}$-values of individual GRKs. Taken together, we speculate that more than one GRK is needed to phosphorylate multiple receptor sites. In line with this, β-arrestin2 was found to need both phosphorylation clusters, whereas β-arrestin1 can be recruited to full extent with distal phosphorylation only (Fig. 4a).

Moreover, the data shown in Fig. 4d, f confirm that GRK phosphorylation is strictly required for the formation of 'hanging' complexes, as neither β-arrestin-dFLR mutant showed recruitment in the absence of GRKs (Supplementary Fig. 7d, f). This experimental setup further strengthens our initial statement that β-arrestin1 is better suited to form a 'hanging' complex with the PTH1R-WT and -PD1 receptor variants in presence of GRKs, compared to β-arrestin2. Additionally, we found that the distal phosphorylation cluster is crucial for the formation of these complexes for both β-arrestins, since the recruitment of the dFLR mutants to PTH1R-PD2 is almost abolished for both isoforms.

## The phosphorylation state of a GPCR induces specific conformational rearrangements in β-arrestin1 and 2

Analogous to the WT receptor, we investigated β-arrestin conformational changes for the PTH1R-PD1 and -PD2 mutants (Supplementary Fig. 3) as well as for the PTH1R WT in ΔQ-GRK (Supplementary Fig. 4). Therefore, we were able to analyse differences in molecular rearrangement between β-arrestin1 and 2 regarding specific C-terminal GPCR phosphorylation (Fig. 5). Our approach enabled us to differentiate between conformational changes that are induced by the active and phosphorylated receptor (P-R*) and the active receptor in the absence of GRK-mediated phosphorylation (R*). Moreover, the use of both cluster mutants aided to further match phosphorylation-dependent effects with the availability of the proximal phosphorylation cluster (deleted in P-R*-PD1), or distal phosphorylation cluster (deleted in P-R*-PD2). Hence, we present a comparison between the phosphorylation-dependent conformational change fingerprints of β-arrestin1 and 2, sorted into sensors located in the N- (Fig. 5a) and C-domain (Fig. 5c) of the two isoforms. A radar chart representation of these data, normalised to the P-R* induced value for each individual biosensor, is shown in Fig. 5b. These phosphorylation-dependent conformational change signatures are also depicted as heat-maps in Supplementary Fig. 9. This representation enables the at-a-glance comparison of phosphorylation-dependency for each biosensor (normalised to each biosensor, Supplementary Fig. 9a), or the complete conformational fingerprints of β-arrestin1 and 2 when coupled to each of the four phosphorylation-specific receptor states (raw Δ net BRET changes, Supplementary Fig. 9b). All conformational change measurements have been subjected to statistical analysis. The results of the comparison between ligand and vehicle stimulation for all biosensors and all phosphorylation-specific conditions can be accessed

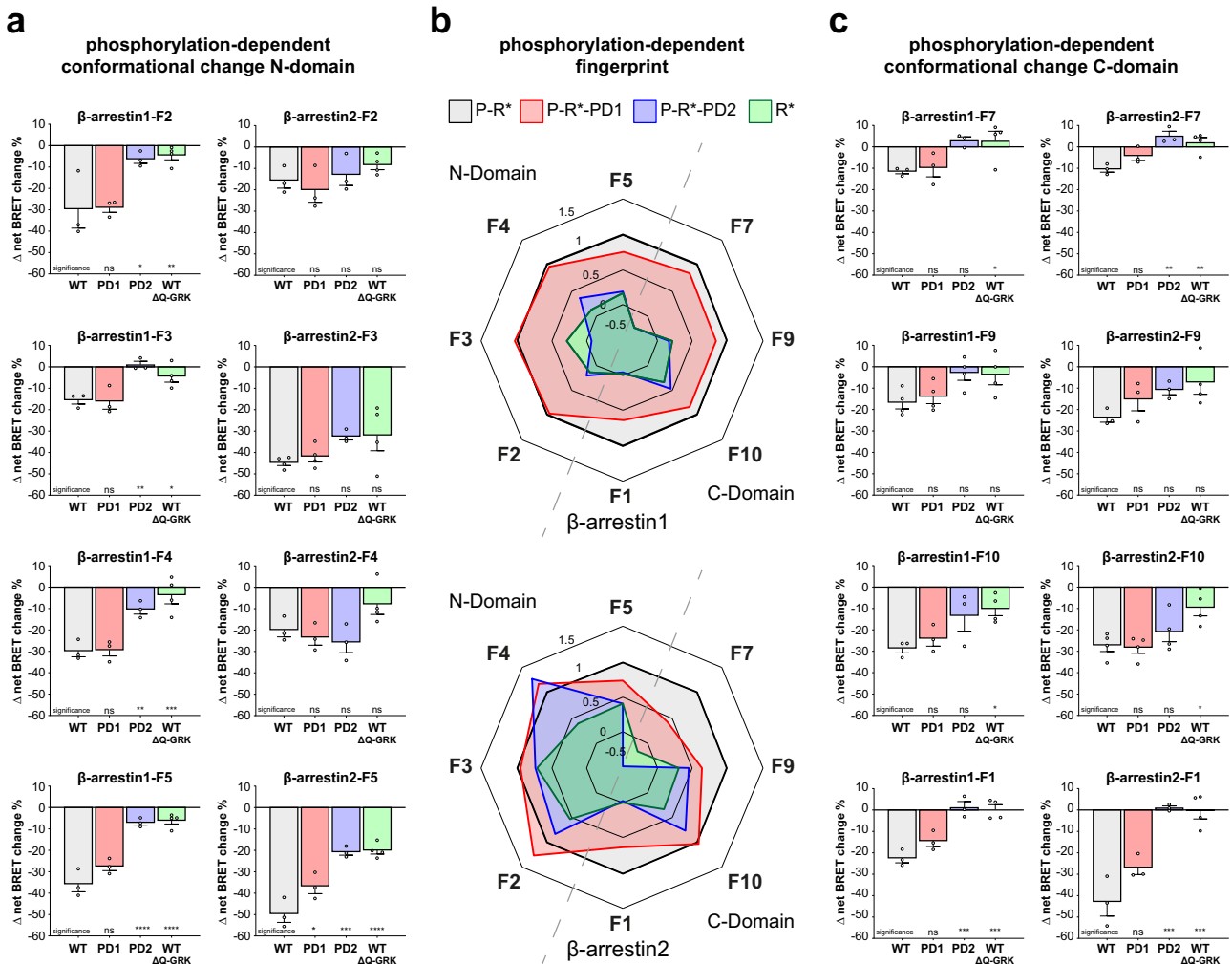

**Fig. 5 | The PTH1R phosphorylation state induces specific conformational rearrangements in β-arrestin1 and 2. a** Conformational change of β-arrestin1 and 2 biosensors labelled in the respective N-domain, interacting with PTH1R-WT, PTH1R-PD1, PTH1R-PD2 in HEK-WT, or PTH1R-WT in ΔQ-GRK. In short, HEK-WT or ΔQ-GRK cells were transfected with either an untagged PTH1R-WT, PTH1R-PD1, or PTH1R-PD2 expression construct and one β-arrestin conformational change biosensor, FlAsH-labelled and stimulated with 3 μM PTH(1-34). Conformational change data are shown as Δ net BRET change in percent, mean of at least three independent repetitions (all measurements in ΔQ-GRK $n = 4$; β-arrestin2-F3 PTH1R-WT, -PD1 $n = 4$; all other conditions $n = 3$) + SEM. The statistical significance was calculated by one-way ANOVA, followed by a two-sided Dunnett's test (*, $p < 0.05$; **, $p < 0.01$; ***, $p < 0.001$; ****, $p < 0.0001$). The complete results of the statistical analysis are

combined in Supplementary Table 5. **b** Shows a radar chart representation of the data from a and c. The data for each individual sensor were normalised to the respective conformational change value induced by PTH1R-WT in each dataset. **c** Shows conformational change measurements of β-arrestin1 and 2 sensors located in the respective C-domains, analogous to the N-domain data presented in a. Results are shown as Δ net BRET change in percent, mean of at least three independent repetitions (all measurements in ΔQ-GRK $n = 4$; β-arrestin1-F9 PTH1R-WT, -PD1, -PD2 $n = 4$; β-arrestin2-F10 PTH1R-WT, -PD1, -PD2 $n = 4$; all other conditions $n = 3$) + SEM. The statistical significance was calculated by one-way ANOVA, followed by a two-sided Dunnett's test (*, $p < 0.05$; **, $p < 0.01$; ***, $p < 0.001$; ****, $p < 0.0001$). The complete results of the statistical analysis are combined in Supplementary Table 5. Source data are provided as a source data file.

in Supplementary Table 4. The statistical comparison between the phosphorylation-specific conditions for each respective biosensor is combined in Supplementary Table 5.

For most sensors, changes in the receptor phosphorylation state altered the measured conformational change signals. Interestingly, the biosensors exhibited different phosphorylation-dependent behaviours, depending on the localisation of the specific FlAsH acceptor site.

In general, the P-R*-PD1-induced conformational change fingerprint of β-arrestin1 is not significantly different to the P-R*-induced fingerprint (Fig. 5). This is in line with the unchanged recruitment of β-arrestin1 to the PTH1R-WT and -PD1 receptor variants (Fig. 4a). In comparison, the β-arrestin2 conformational change sensors exhibited a partially different phosphorylation-dependent behaviour. For β-arrestin1 the F2, F3 and F4 biosensors, which are located in the phosphorylation-sensing N-domain, show the same signal intensity for P-R* and P-R*-PD1, whereas for β-arrestin2, the F2 and F4 sites even

show a slight signal increase for P-R*-PD1. Additionally, we observed that the β-arrestin1 sensors located in the C-domain show a slight signal attenuation for P-R*-PD1. This loss is more pronounced for the β-arrestin2 F7 and F9 sensors, while the measured conformational change of the F10 position stays unaltered. P-R*-PD1-induced conformational changes measured for the F1 and F5 positions show a similar signal reduction for both β-arrestin isoforms.

In contrast, the P-R*-PD2 and R* receptor states induced similar, yet substantially lower conformational changes for all β-arrestin1 biosensors in comparison to the P-R* condition (less than 50% of the P-R*-induced signals in all cases). The observation that the R* receptor state globally mediates a comparable pattern of conformational changes as P-R*-PD2 further supports the notion that the distal PTH1R phosphorylation cluster (not available in P-R*-PD2, depicted in Supplementary Fig. 6) is essential for β-arrestin1 to adopt an active (PTH1R-WT-like) conformation.

Intriguingly, the P-R*-PD2 and R* receptor states still induced a higher degree of conformational changes for β-arrestin2 in comparison to β-arrestin1 (Fig. 5b). Unexpectedly, but interestingly the β-arrestin2-F2 sensor reported similar conformational changes, independent of the receptor phosphorylation state. Strikingly, we were also able to record robust conformational changes of the β-arrestin2-F3 sensor for all conditions in the range of −35 to −45 % Δ net BRET change, maintaining a majority of the signal even in the absence of GRK-mediated receptor phosphorylation (Fig. 5a). In line with the distinct recruitment of β-arrestin2 to the PTH1R-PD1 and -PD2 mutants (Fig. 4a), we conclude that the P-R*-PD1, P-R*-PD2 and R* receptor states induce different active β-arrestin2 conformations in comparison to the fully phosphorylated and activated receptor. Yet, some conformational change biosensors retain a high signal intensity even in the absence of GRK-mediated PTH1R phosphorylation. We attribute these findings to the higher flexibility and lower requirement for C-terminal GPCR phosphorylation of β-arrestin2 in comparison to β-arrestin1.

Although positions F1 and F5 show different conformational change signals for β-arrestin1 and 2 when coupling to P-R* (Fig. 3), they exhibit a similar relative dependence on the receptor phosphorylation state (P-R* > P-R*-PD1 > P-R*-PD2, Fig. 5a, c). For the F5 position we see a small signal reduction for both β-arrestins when coupling to P-R*-PD1, which was more pronounced for both P-R*-PD2 and R*. This behaviour was partially expected, as this labelling site is located in the respective phosphate-sensing N-domains. This observation might directly reflect on the missing C-terminal GPCR phosphorylation, as the F5 loop has been shown to interact with phosphorylated GPCR C-termini[7,10,11]. Additionally, it is tempting to speculate that the proximal phosphorylation cluster is important for the disruption of the FLR-lock[31] (close to the F5 labelling position). This would also explain why the β-arrestin-dFLR mutants are similarly recruited by the PTH1R-WT and PTH1R-PD1 (Fig. 4d, f).

For the F1 position, the interaction with P-R*-PD1 reduced the conformational changes for both β-arrestins by relative 40 %, while for P-R*-PD2 and R* they are abolished (Fig. 5a, c). From these results, we conclude that the membrane-anchoring of the β-arrestin C-edge region is altered by differential receptor phosphorylation. One possible explanation could be differentially engaged complex geometry of β-arrestins and the utilised PTH1R receptor variants, which may lead to an incomplete interaction with the plasma membrane or membranous components[12,14,24].

## β-arrestin-supported receptor internalisation and MAPK signalling are modulated by the receptor phosphorylation state

To assess, whether differential PTH1R phosphorylation leads to the induction of specific β-arrestin-mediated downstream functions, we comprehensively investigated the internalisation and early trafficking characteristics of the PTH1R-WT, -PD1 and -PD2 variants (Fig. 6a–f and Supplementary Fig. 10). For this, we conducted BRET assays utilising the energy transfer between NanoLuc-tagged versions of PTH1R-WT, -PD1 and -PD2 and membrane-localised CAAX-YFP (prenylation sequence of KRas, plasma membrane) and FYVE-mNeonGreen (phosphatidylinositol 3-phosphate (PI3P) binding motive of endofin, early endosomes) constructs, as previously described[42–44].

Figure 6a shows the time-dependent BRET response of all three receptor variants for the redistribution within the plasma membrane (change in proximity to CAAX) and the translocation to early endosomes (FYVE). For the PTH1R-WT, we can convincingly follow these processes, as both assays registered a robust BRET change, reaching their respective plateaus ~10 min after ligand addition. As expected, receptor redistribution within the plasma membrane starts immediately after ligand application (Fig. 6a), while endosome delivery (FYVE association) of the receptor shows a slight delay of about 100 s. Similar kinetics can also be observed for the PTH1R-PD1 variant (Fig. 6a), albeit the data show a slight reduction in signal intensity for both assays.

Interestingly, the PTH1R-PD2 mutant only elicits a minimal response in these assays (Fig. 6a), indicating that the distal phosphorylation site is crucial for proper trafficking of the PTH1R.

In order to assess the impact of β-arrestins and GRKs on the internalisation process of the PTH1R we additionally performed measurements using β-arrestin1/2 knockout cells and ΔQ-GRK (Fig. 6c, d). Our data demonstrate that the loss of β-arrestins diminishes the endosomal arrival of all investigated receptor variants (FYVE association) to the same extent as a loss of GRKs (Fig. 6d). Strikingly, the redistribution of PTH1R-WT within the plasma membrane (CAAX assay) is differentially regulated by GRKs and β-arrestins (Fig. 6c). In absence of β-arrestins the measured signal is reduced by ~30% (Supplementary Fig. 10b), while it is almost abolished in the absence of GRKs. In contrast, the PTH1R-PD1 variant exhibited a similar CAAX dissociation behaviour in HEK-WT and β-arrestin1/2 knockout cells (Fig. 6c), while the PTH1R-PD2 mutant exhibits minimal responses, regardless of the availability of GRKs and β-arrestins (Fig. 6c).

From these results, we can conclude that both, GRKs and β-arrestins are crucial for the initial trafficking of GPCRs, yet their functions affect different aspects of this process. Furthermore, we were able to show that the distal phosphorylation cluster of the PTH1R serves as a master regulator for receptor internalisation. Proximal receptor phosphorylation, in contrast, specifically controls the β-arrestin-facilitated reorganisation of receptors at the plasma membrane and significantly enhances the endosomal delivery of PTH1R (Supplementary Fig. 10d). One possible explanation could be that PTH1R-PD1-induced β-arrestin conformational states (Fig. 5) are not able to further promote complex formations with specific effector proteins (e.g. unable to recruit clathrin, AP2 or other effectors to the PTH1R complex). In contrast, loss of distal receptor phosphorylation in PTH1R-PD2 seems to exclusively generate trafficking-incompetent β-arrestin conformational states and additionally leads to a complete loss of GRK-mediated receptor internalisation.

Utilisation of the FYVE-based BRET acceptor also enabled the investigation of β-arrestin recruitment to early endosomes (Fig. 6e, f). We were able to observe robust concentration-dependent endosomal localisation of both β-arrestin isoforms for PTH1R-WT, which is similarly reduced to a minimum for all receptor variants in ΔQ-GRK. Distal receptor phosphorylation again shows to be the crucial factor that enables the formation of endosomal β-arrestin complexes, yet also the PTH1R-PD1 mutant disproportionally reduced recruitment of β-arrestin1 and 2 to early endosomes by ~50% (Supplementary Fig. 10f and h). These effects can also be attributed to changes in β-arrestin conformations, induced by the different receptor phosphorylation states.

In order to investigate the functionality of β-arrestin-facilitated mitogen-activated protein kinase (MAPK) signalling, we additionally analysed the receptor activation-dependent extracellular signal-regulated kinase (ERK1/2) phosphorylation in HEK-WT and β-arrestin1/2 knockout cells. As previously shown[3,4,20], agonist stimulation of the PTH1R-WT leads to increased ERK1/2 phosphorylation (Fig. 6g, h), which is decreased in the absence of β-arrestins. Strikingly, PTH1R-PD2-induced ERK1/2 phosphorylation is diminished to similar levels despite the presence of β-arrestins. The PTH1R-PD1 displayed no significant reduction compared to PTH1R-WT (Fig. 6d). This is in line with the results of Lee et al., as they proposed that the receptor-dependent ERK1/2 phosphorylation is reflected by conformational changes measured for β-arrestin2-F10[20] (Lee et al. homologous position F5). Consistent with this observation, our β-arrestin conformational change signatures for PTH1R-WT and PTH1R-PD1 showed similar values for the F10 position, while PTH1R-PD2 showed a reduced signal (Fig. 5). The attenuated ERK1/2 signalling behaviour of the PTH1R-PD2 receptor variant might already be explained by its reduced capability to recruit β-arrestin adaptors (Fig. 4a). However, these data could also

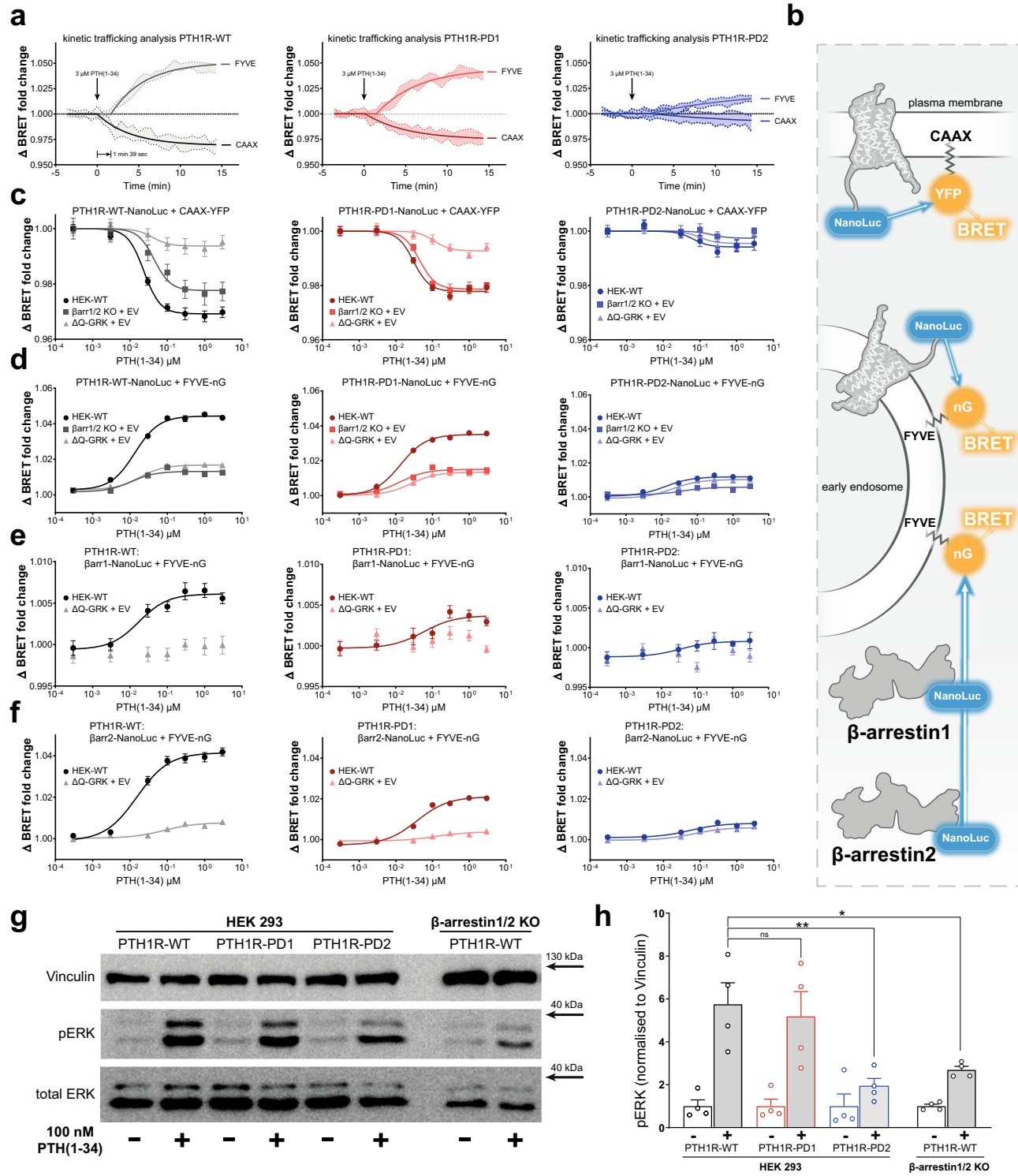

indicate that specifically the phosphorylation of the distal cluster plays an important role in the induction of β-arrestin conformational states that support ERK1/2 signalling. Moreover, when interpreting these results in the context of our comprehensive internalisation analysis, it is tempting to speculate that the reduced capacity of the PTH1R-PD2 variant to localise to early endosomes might be directly responsible for its inability to elicit substantial MAPK signalling. Taken together, our downstream analysis provides clear-cut evidence that GRKs and β-arrestins are crucial effectors that govern the intracellular fate and signalling of the PTH1R, via the induction of specific β-arrestin conformational states.

## Discussion

As universal adaptor proteins modulating distinct signalling outcomes of GPCRs, β-arrestins have been shown to undergo different conformational changes when binding specific receptors or phosphopeptides[10,19–21,32]. Studies also link these GPCR-specific conformational changes to distinct downstream signalling functions, making the point that β-arrestins adjust their functionality according to the geometry of the resulting GPCR–β-arrestin complex. Until now, these assessments of β-arrestin conformational changes mostly focus on one of the two β-arrestin isoforms, making it impossible to judge whether β-arrestin1 and 2 perform redundant, overlapping or

**Fig. 6 | Phosphorylation states of the receptor determine PTH1R internalisation and ERK1/2 signalling. a** Time-dependent BRET change between a NanoLuc-tagged receptor and CAAX-YFP or FYVE-mNeonGreen (nG), measured for PTH1R-WT, -PD1 and -PD2 in HEK-WT. Results are shown as mean of three independent repetitions ($n = 3$) and 95% confidence intervals (coloured areas). **b** schematic depiction of the utilised measurement systems to comprehensively investigate the initial internalisation and trafficking characteristics of the PTH1R. **c** and **d** Concentration-dependent BRET changes, of all three investigated PTH1R variants in the CAAX association and FYVE dissociation assays, as measured in HEK-WT, β-arrestin1/2 double knockout and ΔQ-GRK cells. Cells were transfected with CAAX-YFP or FYVE-mNeonGreen and NanoLuc-tagged constructs of either PTH1R-WT, -PD1, or -PD2 and stimulated with varying concentrations of PTH(1-34), as indicated. Results are shown as mean of three independent repetitions ($n = 3$) ± SEM. **e** and

**f** Analogous measurements, utilising the energy transfer between NanoLuc-fused β-arrestin constructs in the presence of untagged PTH1R (-variants), in order to assess the endosomal delivery of β-arrestins according to the receptor phosphorylation state. Results are shown as mean of three independent repetitions ($n = 3$) ± SEM. Western blot analysis of the PTH1R-dependent induction of ERK1/2 phosphorylation. HEK-WT and β-arrestin1/2 double knockout cells were transfected with the respective PTH1R receptor variants and lysed before or after 20 min of stimulation with 100 nM PTH(1-34). A representative blot (**g**) as well as the mean quantification of four independent experiments (**h**) ($n = 4$) + SEM is shown. The statistical significance was calculated by one-way ANOVA, followed by Tukey's test (*, $p < 0.05$; **, $p < 0.005$). Exact $p$ values for vehicle vs stimulation: PTH1R-WT, $p = 0.0003$; PTH1R-PD1, $p = 0.0015$; PTH1R-PD2, $p = 0.9578$; PTH1R-WT in β-arrestin1/2 double knockout cells, $p = 0.5378$. Source data are provided as a source data file.

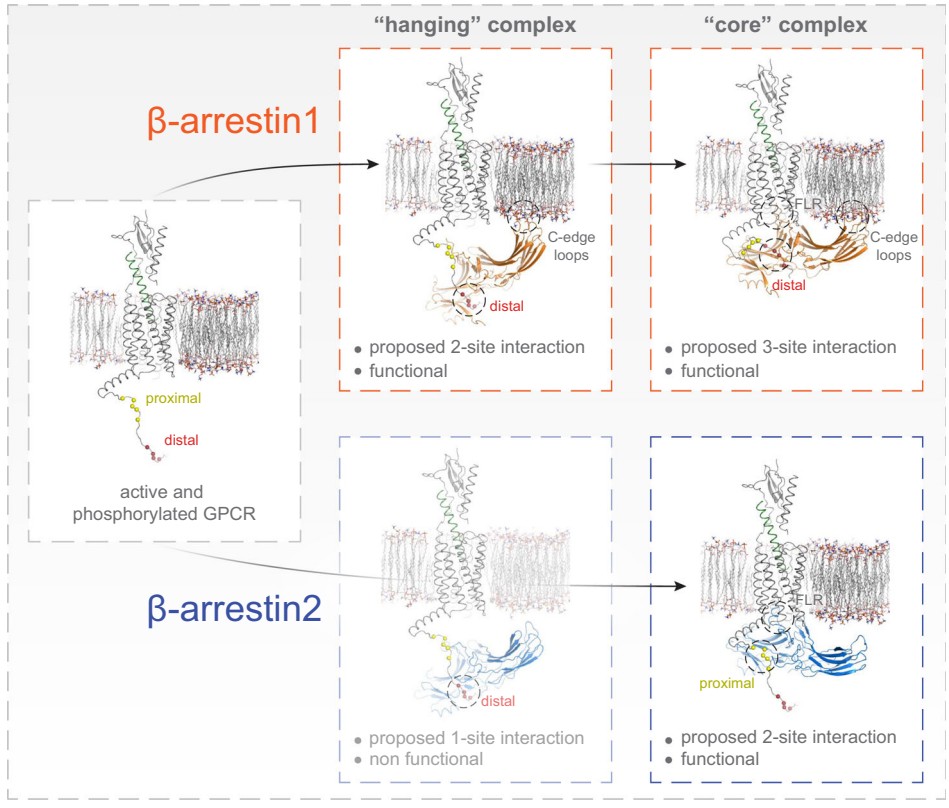

**Fig. 7 | Possible geometry of 'hanging' and 'core' complex configurations between PTH1R and β-arrestin1 and 2.** Molecular modelling of differently formed 'hanging' and 'core' complex configurations between the active and phosphorylated PTH1R (PDB: 6NBF) and β-arrestin1 (PDB: 4JQI) and 2 (PDB: 6K3F), respectively. The results for β-arrestin1 interactions with the PTH1R are depicted in the top row, while the bottom row shows analogous results obtained for β-arrestin2. Our data suggest that β-arrestin1 (top row) interactions with the PTH1R rely more strongly on the distal phosphorylation sites, while the proximal phosphorylation does not seem to stabilise these complexes. Additionally, for β-arrestin1 both 'hanging' and 'core' PTH1R–β-arrestin1 complex configurations are presumably stabilised by membrane anchorage of the exposed C-edge loops[47]. Hence, we propose that PTH1R–β-arrestin1 interactions rely on a 2-site interaction in the

'hanging' configuration, while the additional engagement of the FLR adds a third interaction site in the 'core' configuration. In contrast, for β-arrestin2 the bottom row depicts that the 'hanging' PTH1R–β-arrestin2 complex was characterised with a low stability and showed to be trafficking dysfunctional. This lower preference of β-arrestin2 to form a 'hanging' complex might be explained by its reduced capability to interact with the plasma membrane, as compared to β-arrestin1. This would suggest that PTH1R–β-arrestin2 'hanging' complexes are solely mediated by distal PTH1R phosphorylation, resulting in a less stable 1-site interaction. FLR binding to the PTH1R intracellular cavity rigidifies the complex in the 'core' configuration (2-site interaction, bottom row), while we additionally found that the proximal phosphorylation cluster aids in the stabilisation of the 'core' complex, for β-arrestin2.

divergent functions when binding to same GPCR. It is important to keep in mind that studies working with phosphopeptides and purified arrestins only register one binding interface between β-arrestins and GPCRs, neglecting the impact of the GPCR intracellular cavity on β-arrestin conformational changes[10,21,32].

Along this line, two recent studies proposed that the GPCR intracellular cavity can activate β-arrestins without C tail-mediated interactions[45,46]. To address the above-mentioned experimental

shortcomings, our study combines BRET-based arrestin sensors, arrestins devoid of the finger-loop region (dFLR) and GRK knockout cells. Hence, our approach allows us to comprehensively study conformational changes in β-arrestin1 and 2 in living cells and to separate the contribution of arrestin-binding to the non-phosphorylated active GPCR (R*) via the intracellular cavity or the active GPCR with different degrees of phosphorylation (P-R*). With this broad approach we were able to reveal major differences between the two β-arrestin isoforms,

as we show that they prefer distinct, functional complex configurations (Fig. 1) when binding to the same receptor, resulting in clearly different conformational changes (Fig. 3). Since this characterisation relies on BRET experiments, we can follow β-arrestin conformational changes globally in different conditions to rapidly gain insight about the importance of binding interfaces and general determinants of GPCR–β-arrestin complex formation (Fig. 5). Yet, these sensors are not suitable to provide us with a high molecular resolution on an amino acid level. Nevertheless, we made the attempt to provide a structural interpretation of our results and therefore performed molecular modelling (see methods section for details), informed by the experimental data presented in this study. Here, we used the structure of active PTH1R (PDB:6NBF), as well as β-arrestin1 (PDB: 4JQI) and β-arrestin2 (PDB: 6K3F), respectively, to predict the configuration of differently formed 'hanging' and 'core' complexes (Fig. 7). In conjunction with our experimental data, the modelling results show that β-arrestin1 interactions with the PTH1R rely mostly on distal phosphorylation sites, while proximal phosphorylation does not seem to stabilise these complexes (Fig. 7, top row; Figs. 4a–d and 5). In addition, both 'hanging' and 'core' PTH1R–β-arrestin1 complexes are presumably stabilised by β-arrestin1 membrane interactions via its C-edge loops[47] (Fig. 7, top row; Figs. 3 and 5). Hence, we were able to estimate that PTH1R–β-arrestin1 complex formation relies on a 2-site interaction in the 'hanging' configuration, while binding of the FLR additionally stabilises the complex with a third interaction site in the 'core' configuration (Fig. 7, top row). In comparison to β-arrestin1, the 'hanging' complex formed between the PTH1R and β-arrestin2 showed a lower stability and was not able to facilitate PTH1R internalisation (Fig. 1 and Supplementary Fig. 1b). Considering the differential conformational change signals measured for the C-edge localised β-arrestin1 and 2 F1 sensors (Fig. 3), this lower preference of β-arrestin2 to form a 'hanging' complex might be explained by its reduced capability to interact with the plasma membrane. Following this hypothesis, we propose that that PTH1R–β-arrestin2 'hanging' complexes are solely mediated by distal PTH1R phosphorylation, resulting in a less stable 1-site interaction (Fig. 7, bottom row; Fig. 4f). Nonetheless, binding of the β-arrestin2 FLR to the PTH1R intracellular cavity is able to rigidify the complex in the 'core' configuration (2-site interaction). Moreover, we additionally found that proximal PTH1R phosphorylation aids in the stabilisation of the β-arrestin2 'core' complex, in contrast to β-arrestin1 (Fig. 7, bottom row; Fig. 4a). We speculate that these different complex configurations exist in a certain equilibrium, yet we cannot exclude that 'hanging' and 'core' configurations are formed consecutively as part of a multi-step binding mechanism, as recently suggest by dynamic modelling[48]. Ultimately, our data would allow for the interpretation that β-arrestin1 is better suited to form a 'hanging' complex, whereas β-arrestin2 binding relies more on the interaction with the intracellular GPCR cavity (Figs. 1, 4 and Supplementary Fig. 1b). A similar behaviour was recently described for another class B GPCR (as defined by ref. 49), specifically the vasopressin 2 receptor[29]. This constitutes a major difference between the two isoforms, especially since β-arrestin1 can still drive receptor internalisation without the FLR, whereas β-arrestin2 fails to form a functional 'hanging' complex (Fig. 1h, i and Supplementary Fig. 1).

This finding might be directly reflected by several reports of β-arrestin1 engaging phosphorylated peptide stretches of non-GPCR signalling molecules, like e.g. receptor tyrosine kinases[50,51]. Furthermore, we located the hotspots of differential conformational changes in the phosphate-sensing N-domains and C-edge regions of the two proteins (Fig. 3a, c), suggesting that they interact differently with the phosphorylated C-terminus of a GPCR and the cell membrane. Z-factor analysis of the newly designed β-arrestin-F5 sensors revealed that this measuring system is suitable for high-throughput screening (Supplementary Fig. 5). This might prove especially useful, as measurements featuring these sensors require no modification of the tested GPCR

and could potentially constitute a screening platform for e.g. the deorphanisation of receptors or the development of biased agonists.

As our measurements comprise not only conformational changes induced by the PTH1R-WT but also utilise two cluster mutants, missing key C-terminal phosphorylation sites, we provide in cellulo experimental proof of the computational and biochemical results of refs. 21,32, confirming that β-arrestin conformational changes are, in fact, dependent on the specific C-terminal phosphorylation of a GPCR. Furthermore, we recorded the complete conformational fingerprint of both β-arrestins when coupling to the PTH1R in the absence of GRKs, using quadruple GRK knockout cells (Fig. 5 and Supplementary Fig. 4). As all three tested receptor variants induced identical β-arrestin recruitment in ΔQ-GRK (Fig. 4c, e and Supplementary Fig. 7), we assume that no other intracellular kinases play a decisive role in this process, suggesting that the assessed receptor variants are unphosphorylated in this condition. This constitutes a major advancement over previous studies[10,19,20,32], as we are able to differentiate phosphorylation-dependent β-arrestin conformational changes from the ones that are induced by the intracellular cavity of a GPCR, in living cells. Strikingly, we found positions in β-arrestin2 that undergo similar conformational changes independent of receptor phosphorylation (Fig. 5b). This is again in line with the finding, that β-arrestin2 functions are more reliant on the formation of a 'core' complex, especially since all tested sites for β-arrestin1 are sensitive towards differential GPCR phosphorylation (Fig. 5).

In addition, our β-arrestin conformational change biosensors might provide further insight into GPCR–arrestin complex configurations. For coupling with the PTH1R-WT, the biosensors labelled in the F3 position showed reduced values for β-arrestin1 in comparison to β-arrestin2 (Fig. 3b). Corresponding residues in the F3 loop of visual arrestin were reported to respond differently when binding to phosphorylated, light-activated Rhodopsin (P-R*)[52] or phosphorylated Opsin (P-R)[53]. Since phosphorylated Opsin is in a predominantly inactive conformation one might assume that it is unable to form a 'core' complex with arrestin, interacting mostly in a 'hanging' complex configuration. Therefore, it is tempting to speculate that the F3 labelling position might be sensitive to the formation of distinct GPCR–arrestin complex configurations and could be able to differentiate between 'hanging' and 'core' complexes. The retained, phosphorylation-independent conformational changes for β-arrestin2-F3 (Fig. 5a, b, Supplementary Fig. 3b and Supplementary Fig. 4b) possibly result from a virtually unaltered capability to form a 'core' complex with the PTH1R, regardless of the receptor phosphorylation state. Conformational changes measured for the F3 position in β-arrestin1 were found to be specifically induced by the distal phosphorylation cluster (Fig. 5a). This would be in line with the hypothesis that proximal receptor phosphorylation is dispensable for the formation of a 'hanging' complex and distal phosphorylation sites, still present in PTH1R-PD1, predominantly enable the formation of such complexes[31].

Following this hypothesis, we deleted the FLR of β-arrestin1-F5, a sensor which showed different conformational changes for the P-R* and P-R*-PD1. We conclude that these differences in conformational change are exclusively induced by the proximal phosphorylation cluster and dependent on the formation of a tight 'core' complex. Thus, removal of the FLR should abolish this difference, as the dFLR mutant should only be able to associate with the receptor in a 'hanging' complex. Indeed, the β-arrestin1 mutant biosensor changed its behaviour and registered almost identical signals for the interaction with P-R* and P-R*-PD1 upon deletion of the FLR, as it cannot make use of the 'core' complex binding interface (Supplementary Fig. 11). The PTH1R-PD2 mutant did not induce concentration-dependent conformational changes at the β-arrestin1-dFLR-F5 sensor. Again, this would be in line with the distal phosphorylation cluster being a crucial binding interface for the formation of a 'hanging' PTH1R–β-arrestin complex.

Our analysis of β-arrestin-mediated downstream functions in Fig. 6 links the phosphorylation state of the PTH1R to distinct consequences regarding the intracellular shuttling and signalling of active receptor molecules. Here, it is important to note that the PTH1R is able to activate $G_s$ at the plasma membrane in order to invoke an initial cAMP response. Interestingly, this signalling persists following β-arrestin-facilitated receptor endocytosis[27]. However, the PTH1R also elicits physiologically important[54] $G_q$ signalling upon agonist-binding. In contrast to the prolonged, intracellular $G_s$ signalling of the receptor, PTH1R-mediated activation of $G_q$ is a transient process that exclusively occurs at the plasma membrane[55]. We show that specific β-arrestin conformational states influence the internalisation characteristics of PTH1R, thus it is tempting to speculate that β-arrestins are responsible for this 'switch' in receptor-mediated signalling. Following these arguments, there is now surmounting evidence that β-arrestins play an important role in the regulation and balancing of different signalling events via the formation of specific complex configurations. Still further investigation is needed to specifically link the functionality of β-arrestins with the spatio-temporal characteristics of primary GPCR signalling.

In summary, our findings demonstrate inherent differences between the two homologous β-arrestin isoforms for the interaction with the same GPCR. Furthermore, we show that the phosphorylation state of a given receptor induces specific conformational rearrangements that determine the functional diversity between the two β-arrestin isoforms.

## Methods

### Molecular cloning and construct origin

β-arrestin2 conformational change biosensors were created on the basis of constructs described in ref. 19. To enable BRET measurements, the CFP-tag was exchanged with the NanoLuc gene (Promega). For FlAsH insertions at positions F9 and F10 we used the homologous positions as described in ref. 20 (their positions F4, F5, respectively). β-arrestin1 constructs were designed homologously. A table of insertion sites for FlAsH-binding (CCPGCC) for all utilised β-arrestin conformational change biosensors is shown below (Table 1).

The C-terminal tags of PTH1R constructs previously described in Zindel et al. 2016[40] were exchanged with the Halo-tag gene (Promega). The Rab5-mCherry construct was kindly provided by Tom Kirchhausen

(Harvard Medical School, Boston, USA). Origin of GRK constructs is described in ref. 29.

β-arrestin-dFLR constructs were designed according to ref. 9 by site-directed mutagenesis. For β-arrestin1 we deleted amino acids Y63 to K77 and for β-arrestin2 we deleted amino acids Y64 to K78. Primers used are shown in table below (Table 2).

### Cell culture

HEK293 cells (HEK-WT) were originally obtained from the DSMZ Germany (ACC 305). The HEK293 cell knockout derivatives were either created in-house (ΔQ-GRK cells[29]) or provided by co-authors (β-arrestin1/2 double knockout cells[36]). All cell lines were cultured at 37 °C with 5% $CO_2$ in Dulbecco's modified Eagle's medium (DMEM; Sigma-Aldrich #D6429), supplemented with 10% foetal calf serum (FCS; Sigma-Aldrich #F7524) and 1% penicillin and streptomycin (Sigma-Aldrich #P0781). The cells were passaged every 3–4 days and seeded to achieve a confluency between 70 and 90% for experiments relying on transient transfection. All cell lines were regularly checked for mycoplasma infection using the Lonza MycoAlert mycoplasma detection kit (LT07-318) and were found to be negative.

### Intermolecular bioluminescence resonance energy transfer (BRET)

The β-arrestin recruitment experiments were performed in HEK293 or ΔQ-GRK cells[29]. The cells were seeded in 21 cm² dishes, transfected with 1.5 µg PTH1R C-terminally coupled to a Halo-ligand binding Halo-tag and 0.375 µg of β-arrestin C-terminally fused to a Nanoluciferase (NanoLuc), according to the Effectene transfection reagent manual by Qiagen. After 24 h, we seeded 40,000 cells per well into poly-D-lysine coated 96-well plates (Brand, #781965). For labelling of the receptor coupled Halo-tag, Halo-ligand(618) (Promega, #G980A) was added 1:2,000 to the cell suspension. For each transfection, triplicates and a condition lacking the Halo-ligand(618) (mock labelling) were seeded. Another 24 h later, the cells were washed twice with measuring buffer (140 mM NaCl, 10 mM HEPES, 5.4 mM KCl, 2 mM $CaCl_2$, 1 mM $MgCl_2$; pH 7.3) and NanoLuc substrate furimazine (Promega, #N157A) was added 1:35,000 in measuring buffer. The concentration-dependent BRET change was measured using a Synergy Neo2 plate reader (Biotek; Gen5 software), containing a custom-made filter (excitation bandwidth 541-550 nm, emission 560-595 nm, fluorescence filter 620/15 nm). The baseline measurements were conducted for three minutes. After stimulation with the respective PTH(1-34) concentrations, the measurements were continued for five minutes and all data points recorded two minutes post ligand addition were subsequently averaged. Initial BRET changes were calculated by the division of the stimulated values by baseline values. The initial BRET change was then corrected for labelling efficiency via subtraction of values generated by mock labelling. To achieve the final Δ net BRET change, the corrected BRET change was divided by the vehicle control. The recruitment of the different β-arrestin-FlAsH conformational change sensors and of the β-arrestin-dFLR constructs to the PTH1R was measured accordingly.

Analysis of PTH1R internalisation characteristics and β-arrestin recruitment to early endosomes was conducted using NanoLuc-tagged versions of PTH1R-WT, -PD1, -PD2 and β-arrestin1 or 2, as well as membrane-tethered genetic fluorophores in the form of CAAX-YFP[42] and FYVE-mNeonGreen[42–44]. HEK293, β-arrestin1/2 knockout and ΔQ-

## Table 1 | Insertion sites for FlAsH-binding

|  | insertion site | |
| --- | --- | --- |
|  | β-arrestin1 | β-arrestin2 |
| F1 | 330 | 331 |
| F2 | 153 | 154 |
| F3 | 48 | 49 |
| F4 | 149 | 150 |
| F5 | 156 | 157 |
| F6 | 325 | 326 |
| F7 | 342 | 335 |
| F8 | 192 | 193 |
| F9 | 224 | 225 |
| F10 | 262 | 263 |

## Table 2 | Primers used to design β-arrestin-dFLR constructs

| Primer | DNA Sequence |
| --- | --- |
| βArr1-dFLR_fwd | CTG ACC TGC GCC TTC CGC GAC CTG TTT GTG GCC AAC G |
| βArr1-dFLR_rev | GTT GGC CAC AAA CAG GTC GCG GAA GGC GCA GGT CAG CG |
| βArr2-dFLR_fwd | CTC ACC TGC GCC TTT CGC GAC CTG TTC ATC GCC AAC TAC C |
| βArr2-dFLR_rev | GTT GGC GAT GAA CAG GTC GCG AAA GGC GCA GGT GAG GG |

GRK cells were transfected with both partners of the respective chosen BRET pair in a ratio of 1:10 (BRET donor to acceptor). Due to the utilised chromophores, these measurements were possible without preceding labelling of the cells. The measurements were conducted and analysed analogously to the β-arrestin recruitment assays, but prolonged to 15 min after ligand application.

## Confocal microscopy

For live cell microscopy, β-arrestin1/2 double knockout cells and ΔQ-GRK were seeded into 21 cm² dishes and transfected with 1 μg of PTH1R-CFP, 0.5 μg of β-arrestin-YFP and 0.5 μg of Rab5-mCherry, according to the Effectene transfection reagent manual by Qiagen. After 24 h, 700,000 cells were seeded onto poly-D-lysine coated glass cover slips (three per six-well) and were ready for microscopy the following day.

Before microscopy, the cells were washed twice with measuring buffer. The confocal microscopy images were acquired before and 15 min after stimulation with 100 nM PTH(1-34) at the Leica SP8 laser scanning microscope (Leica Application Suite X software) in a 1024 × 1024 pixel format, using a 63x water immersion objective, zoom factor 3, line average 3 and 400 Hz. CFP was excited at 442 nm, mCherry at 561 nm and YFP at 514 nm. The features of the acquired confocal images were segmented and quantified using an ImageJ based software called segmentation and quantification of subcellular shapes (Squassh)[56]. After Squassh's deconvolution, denoising and segmentation of the two or three fluorescence channels of each image, the raw data readout was eligible for analysis using the R based software SquasshAnalyst[57] as described by A. Rizk. All image derived data in this study was processed and analysed with this method.

## Western blot

For analysis of ERK phosphorylation, 600,000 HEK293 cells or β-arrestin1/2 KO cells were seeded in six-well plates 24 h before transfection. The cells were transfected with 4 μg of indicated expression plasmids using PEI (Sigma-Aldrich, 408727, diluted to 10 μg/ml, pH 7.2 adjusted with HCl). After 24 h, the cells were starved from FCS for 4 h and cells were then treated with 100 nM PTH(1-34) for 20 min or left untreated. Cells were then washed with ice-cold PBS and lysed with RIPA lysis buffer (1% NP-40, 1 mM EDTA, 50 mM Tris pH 7.4, 150 mM NaCl, 0.25% sodium deoxycholate), supplemented with protease and phosphatase inhibitor cocktails (Roche, #04693132001, #04906845001). Samples were run on polyacrylamide gels and analysed for vinculin (BIOZOL, BZL03106; 1:1000), pERK (phospho-p44/42, Cell signaling technology, #9106; 1:1000) or total ERK (p44/42, cell signaling technology, #9107; 1:1000) as indicated. For primary antibody detection goat anti-rabbit (SeraCare, #5220-0336; 1:10,000) and goat anti-mouse (SeraCare, #5220-0341; 1:10000) were used. Quantification was done using Fujifilm Multi Gauge V3.0 software.

## Intramolecular BRET

HEK293 cells were seeded into 21 cm² dishes and transfected with 1.2 μg untagged receptor, 0.12 μg of the respective β-arrestin FlAsH-tagged biosensor C-terminally coupled to NanoLuc and empty vector to adjust the total amount of DNA to 2 μg, following the Effectene transfection reagent protocol by Qiagen. 24 h after transfection, we seeded 40,000 cells per well into poly-D-lysine coated 96-well plates and incubated overnight at 37 °C. For this study, the FlAsH-labelling procedure previously described by ref. [58] was adjusted for 96-well plates. Before the FlAsH-labelling procedure, the cells were washed twice with PBS, then incubated with 250 nM FlAsH in labelling buffer (150 mM NaCl, 10 mM HEPES, 25 mM KCl, 4 mM CaCl₂, 2 mM MgCl₂, 10 mM glucose; pH7.3), complemented with 12.5 μM 1,2-ethane dithiol (EDT) for 60 min at 37 °C. After aspiration of the FlAsH labelling or mock labelling solutions, the cells were incubated for 10 min at 37 °C with 100 μl of 250 μM EDT in labelling buffer per well. Addition of the

NanoLuc substrate, measurement and analysis of the BRET change was performed as described above (see intermolecular BRET).

## Statistical analysis and evaluation of Z-factor

BRET ratios and fold changes are displayed as mean of at least three independent experiments with error bars indicating the SEM. Statistical analysis was performed using Student's *t* test or analysis of variance (ANOVA; one-way or two-way ANOVA) as well as appropriate multiple comparisons as indicated in corresponding figure legends. Statistical analysis was conducted using Graphpad Prism 7. A type I error probability of 0.05 was considered to be significant in all cases.

Z-factors for β-arrestin1/2-F5 conformational change biosensors were assessed by utilising data of time-dependent signal following stimulation of the PTH1R with 3 μM PTH(1-34). Calculations were conducted in Microsoft Office Excel following the original publication of ref. [38]. To assess means ($\mu$) and standard deviations ($\sigma$), individual data points recorded after 200 s of stimulation with either vehicle (*c*) or 3 μM PTH(1-34) (*s*) were used and applied in Eq. [(1)]:

$$Z = 1 - \frac{(3\sigma_s + 3\sigma_c)}{|\mu_s - \mu_c|} \qquad (1)$$

## Molecular modelling

The cryo-EM structure of active PTH1R in complex with Gs (PDB: 6NBF)[59], the crystal structure of active β-arrestin 1 bound to a vaso-pressin V2 receptor phosphopeptide (PDB: 4JQI)[7] and the crystal structure of active β-arrestin 2 bound to a chemokine CXCR7 receptor phosphopeptide (PDB: 6K3F)[41] were used as structural templates to build the models of the complexes of PTH1R with β-arrestin 1 and 2.

To model the 'hanging' complex configurations between the receptor and β-arrestin1 and 2, we first built the PTH1R C-terminal residues 482-510 (which are missing in the template 6NBF) in an extended conformation using molecular graphics software (The PyMOL Molecular Graphics System, Version 2.5.2 Schrödinger, LLC). This software was then used to manually place the structures of β-arrestin 1 and 2 in a pose that allows the interaction between the distal phosphorylation site of the receptor and the polar core of β-arrestin1 and 2.

The extended conformation of the PTH1R C-terminus was also used to build the 'core' complex with β-arrestin 2, as it allows the interaction between the proximal phosphorylation site of the receptor and the respective polar core. However, such extended conformation of the receptor C-terminus does not allow the formation of a β-arrestin 1 complex compatible with a 3-site interaction involving the polar core, the FLR, and the C-edge loops. Thus, the software Modeller[60] was used to build an alternative conformation of the PTH1R C-terminal residues 482-510 that allows the interaction between the distal phosphorylation site and the polar core of the β-arrestin 1 while maintaining the rest of the expected interactions in the 'core' complex.

## Reporting summary

Further information on research design is available in the Nature Research Reporting Summary linked to this article.

## Data availability

All source data are provided in this paper. The following protein structures were used in this manuscript, accessed via the Protein Data Bank: 4JQI 6K3F 6NBF 1CF1 5W0P 3P2D 2WTR. Source data are provided in this paper.

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

## Acknowledgements

We thank Dr. Aurélien Rizk for his assistance in the evaluation of co-localisation in confocal image datasets. This research was supported by the European Regional Development Fund (Grant ID: EFRE HSB 2018 0019), the federal state of Thuringia and the Deutsche Forschungsgemeinschaft (Grant CRC166 ReceptorLight project C02). J.D. is additionally funded by the University Hospital Jena IZKF (Grant ID: MSP10). A.I. was funded by the LEAP JP19gm0010004 from the Japan Agency for Medical Research and Development (AMED). A.C. was supported by the Luxembourg National Research Fund (INTER/FWO 'Nanokine' grant 15/10358798, INTER/FNRS grants 20/15084569, and PoC 'Megakine' 19/14209621), F.R.S.-FNRS-Télévie (7.8504.20).

## Author contributions

R.S.H., E.S.F.M., J.D. and M.R. conducted, compiled, analysed and visualised the experimental work presented in this study. U.Z. conducted molecular cloning of the NanoLuc/FlAsH biosensors. C.H. and R.S.H. designed the concept of the project. R.S.H. designed all wet-lab experiments. X.D. performed the molecular modelling of different PTH1R–β-arrestin complex configurations. C.K. provided the phosphorylation-deficient PTH1R mutant constructs. A.I. provided the β-arrestin1, 2 double knockout cell line. A.C. provided the FYVE-domain constructs. C.H. supervised and coordinated the project. The manuscript was written by R.S.H., E.S.F.M., J.D. and C.H. with contributions and approval from all listed authors.

## Funding

## Competing interests

The authors declare no competing interests.
