## [Peer Review File · Nature Communications]

β -arrestin1 and 2 exhibit distinct phosphorylation-dependent conformations when coupling to the same GPCR in living cellsReviewer #1 (Remarks to the Author):

The study by Haider et al is aimed at examining the conformation barrestin1 and barrestin2 assume in various complexes with PTH1R. The receptor binds both arrestin isoforms well, but the conformational preferences of arrestin isoforms as well as their requirements for the receptor phosphorylation remain poorly understood. The study employs BRET biosensors as well as a multitude of imaging and other techniques to examine the configuration of the receptor-arrestin complexes and their functional consequences. The study approaches a rather obscure subject shedding light on many details defining the specific features of the interaction of arrestin isoforms with GPCRs. It should be of great interest for the GPCR/arrestin field.

From the conceptual standpoint, it is regrettable that the authors did not attempt to incorporate their conformational data, together with the available structures of the GPCR-b-arrestin1 complexes as well as the studies utilizing biophysical methods, into a structural model. Unfortunately, no structures of such complexes exist for b-arrestin2. Nevertheless, the authors could have capitalized on their finding of the differential conformational changes in these isoforms upon the receptor binding to offer a provisional model for b-arrestin2 in the complex as well. Otherwise, a reader is left with a sense of dissatisfaction of seeing nice diagrams of the sensor responsiveness to the receptor binding but no structural interpretation of these changes and/or differences between the arrestin isoforms.

Another issue is the distinction between the "hanging" and "core" complexes. Although it has been reported that arrestins could form such distinct complexes, it seems that the issue has been somewhat overinterpreted in this case. First, deletion of all GRKs essentially abolished the recruitment of both arrestins to the WT receptor, which suggests that at least this receptor does not form "core" complexes independently of phosphorylation. Second, the defects seen in mutants with the deletion of the finger loop (like in b-arrestin2 in Fig 1c) do not necessarily mean that the arrestin recruitment relies on "both GRK-mediated receptor phosphorylation, as well as the FLR interaction interface" as independent entities. The finger loop has been shown to change conformation upon binding to active phosphorylated GPCR (see, for example, here: J Biol Chem, 2014 Jul 25;289(30):20991-1002; Cell, 2020 Dec 23;183(7):1813-1825). It might be that phosphorylation-induced fit in the finger loop contributes to the high-affinity arrestin binding, and the absence of such contribution might be the reason for the reduced functionality.

The statement that b-arrestin2 engages active GPCR independently of phosphorylation better than b-arrestin1 (page 3 line 159) is not supported by the data presented: in cells lacking GRKs the binding of both isoforms is essentially eliminated (Fig 1c). Furthermore, if anything, b-arrestin2 seems to have higher demands for receptor phosphorylation, since it needs both phosphorylation clusters to be present for the full function (Fig. 4a,e). The authors seem to base this statement on the effect of the finger loop deletion, which is more detrimental in b-arrestin2. However, as mentioned above, this could have alternative interpretations.

Fig 1 c: The description of the statistical analysis is unclear: If Dunnett's test was used, how is the significance between dFLP and WT in ΔQ was arrived at? Dunnette's test compares each treatment to a single control.

The "hanging" conformation was originally proposed as the conformation competent to support receptor internalization and arrestin signaling but not receptor desensitization (Proc Natl Acad Sci U S A. 2017 Mar 7; 114(10): 2562–2567). This does not seem to be the case here, as barr2-dFLP essentially does not support receptor internalization, and the results are unclear for barr1-dFLP: Fig 1f: The difference between barr1-dFLR and no barr1 is not shown as statistically significant. How would the authors interpret that in view of their data?

Fig. 1f,i: There is no significant difference between no-barr and Δ Q-GRK for either b-arrestin1 or b-arrestin2. Therefore, the statement that "other phosphorylation-dependent but b-arrestin-independent ways must exist to internalize the PTH1R" is premature. This statement is contradicted by the authors' own data showing that the removal of either b-arrestins or GRKs has equally detrimental effect of the PTH1R trafficking to endosomes (Fig. 6d).

Figs 1d,g: There does not seem to be quantification for these data, which makes it impossible to compare arrestin recruitment/internalization behavior with the receptor internalization data. Furthermore - and this applies to Figs 1d,g,f, I and Suppl Table 1 - it is unclear how the cells have been selected for the analysis from how many slides, experiments, etc. Additionally, these experiments need the data on the expression levels of all proteins involved, because that would heavily influence the image analysis data, particularly because different modified cell lines are involved.

Suppl Fig 7: GRK expression levels? How the levels compare among different GRK isoforms and between the experiments with b-arrestin1 and b-arrestin2? It seems that no individual GRK is capable of restoring the recruitment of b-arrestin2 to PTR1R-WT, whereas any is sufficient to restore the recruitment of b-arrestin1. However, this could be simply due to inconsistencies in the expression levels.

Fig 6g: Arrestin-dependent ERK activation is diminished when induced via PTH1R-PD2 as compared to the WT receptor. The authors suggest that the reduced capacity of PTH1R-PD2 to localize to endosomes might be directly responsible for this defect. Although arrestin could drive the ERK activation from endosomes, it is probably not always the case. There is strong evidence that only receptor-bound arrestins activate ERK. Thus, a simpler alternative explanation would be that PTH1R-PD2 is less efficient in activating ERK via arrestins because it is less efficient in recruiting arrestins.

Reviewer #2 (Remarks to the Author):

The manuscript by Haider et al. focuses on the two isoforms of β -arrestins (1 and 2), their ability to elicit distinct conformations following their interaction with agonist-stimulated GPCR, in this case, the PTH1R in living cells. In particular, the authors investigated the interplay between specific β -arrestin1/2 conformations and different C-terminus receptor phosphorylation patterns (full, partial at proximal, partial at distal, or altogether none via CRISPR Δ Q-GRK cells). This study extends Cahill III et al.'s work from 2017 (focused on β -arrestin1 isoform only) by investigating the two activation pathways and agonist-activated C-terminus phosphorylation pattern dependence corresponding to 'hanging', and 'core' GPCR- β -arrestin conformations utilizing the two β -arrestin isoforms (1 and 2). The authors employed a combination of BRET-based β -arrestin recruitment, internalization, β -arrestin conformational changes (NanoLuc/FlAsH)-biosensors, confocal microscopy, and signaling downstream of PTH1R (ERK1/2). The results from this study raise exciting opportunities for exploring the phosphorylation barcode for different signaling outcomes 'based signaling'; and the conformational and signaling outputs by the two isoforms of β -arrestin. Overall, the article is well written, the experiments are well designed and rigorously analyzed, and the data broadly support the conclusions. I believe it will be of broad interest to scientists studying GPCR signaling, biased agonism, and GPCR pharmacology. I recommend its publication in Nature Communications after the authors clarify the questions raised below:

- 1. The authors observe distinct conformational signatures on β -arrestins dictated by differentially phosphorylated receptors using the BRET-based conformational biosensors. As well as how these differentially phosphorylated receptor patterns engage**

with WT β -arrestin 1/2 or their finger loop deleted counterparts to deduce the "core" and the "hanging" complexes. While all this is fascinating, it is still not clear (for example, in lines 145-156 and elsewhere throughout the manuscript) what the contribution of each β -arrestin isoform (say percentage) is in inducing particularly the "core" or "hanging" (as in Δ Q-GRK cells) conformation? I feel quantification of these specific conformations is essential in this study. For example, in line 146, the authors indicate they obtained \sim 36% of "hanging" complexes when using β -arrestin1 isoform. Throughout the manuscript, the contribution of a specific isoform to these two conformations, particularly on the 'core' confirmation, wasn't clear. Overall it appears to be minor (or perhaps very difficult to deduce) from these BRET-based data? The authors need to clarify this throughout the manuscript. What would the β -arrestin 1/2 conformational signatures pattern in Fig. 3 be with a different receptor (another class B or just class A)?

2. The conclusion on signaling competence of receptor- β -arrestin conformations on downstream ERK1/2 phosphorylation is only done in the context of the receptor's phosphorylation patterns (WT, PD1, and PD2). The study did not include what may be happening in ERK1/2 signaling in the context of two isoforms of β -arrestin vis-a-vis the phosphorylation patterns. Such data would strengthen the manuscript significantly.

3. Minor: Overall, the figures look great, but some of the labels' font size and color choices (e.g., yellow) are hard to visualize. The authors may need re-formatting the figures and enlarge the labels' relative sizes. I had so much difficulty reading the labels, for example, in Figures: 2, 4, and 6 and in many SI ones. For instance, what EV in the figures, as in 4 implies, wasn't described in the figure legends.

For convenience of the reviewers, we provide a list of changes in display items.

List of changes in display items

- Figure 1:** Data was significantly extended with updated **Supplementary Figure 1**.
- Figure 2:** The yellow label for “F4” was made darker to improve readability.
- Figure 3:** No changes.
- Figure 4:** Label sizes were increased to improve readability.
- Figure 5:** No changes.
- Figure 6:** Label sizes were increased to improve readability.
- New Figure 7:** Molecular modelling that sums up our findings and provides a structural interpretation of differences between PTH1R- β -arrestin complex configurations.
-
- New Supplementary Figure 1:**
- a, b, c** concentration-response curves from **Figure 1b** were sorted according to the probed conditions and are shown again with appropriate y-axis limits to enable evaluation of β -arrestin recruitment in Δ Q-GRK.
 - d, e** the data was replotted and the representation was changed to resemble the layout and design of kinetic traces shown in **Figure 6a**.
 - f, g** quantification of co-localisation between β -arrestin and the PTH1R in Δ Q-GRK was added to the panels.
 - h, i** image statistics concerning the mean object intensity and total object signal per cell size were added to enable evaluation of sample size, as well as transfection, image and segmentation quality (basal images only).
- Supplementary Figure 2:** Label sizes were increased to improve readability.
- Supplementary Figure 3:** No changes.
- Supplementary Figure 4:** No changes.
- Supplementary Figure 5:** No changes.
- Supplementary Figure 6:** No changes.
- Supplementary Figure 7:** Label sizes were increased to improve readability.
- New Supplementary Figure 8:** Additional experiment addressing the homogeneity of GRK expression levels in our GRK-specific β -arrestin recruitment assay.
- Supplementary Figure 9:** Data was previously presented in **Supplementary Figure 8**.
- Supplementary Figure 10:** Data was previously presented in **Supplementary Figure 9**. Label sizes were increased to improve readability.
- Supplementary Figure 11:** Data was previously presented in **Supplementary Figure 10**.

Point-to-point reply to the reviewers' comments:

We thank both referees as well as the editor for their fair and constructive feedback regarding the manuscript at hand. We addressed the reviewers' criticism in the revised version of Haider et al. and included new analyses and experiments to strengthen our line of reasoning. We sincerely believe that the well-balanced suggestions made by the reviewers improved our existing work.

Reviewer #1 (Remarks to the Author):

The study by Haider et al is aimed at examining the conformation of β -arrestin1 and β -arrestin2 in various complexes with PTH1R. The receptor binds both arrestin isoforms well, but the conformational preferences of arrestin isoforms as well as their requirements for the receptor phosphorylation remain poorly understood. The study employs BRET biosensors as well as a multitude of imaging and other techniques to examine the configuration of the receptor-arrestin complexes and their functional consequences. The study approaches a rather obscure subject shedding light on many details defining the specific features of the interaction of arrestin isoforms with GPCRs. It should be of great interest for the GPCR/arrestin field.

We thank the referee for her or his comprehensive summary and fair review of our study. Appreciating the reviewer's interest in our work, we elaborated on the changes that result from the specified suggestions and questions in the point-to-point review below.

- From the conceptual standpoint, it is regrettable that the authors did not attempt to incorporate their conformational data, together with the available structures of the GPCR- β -arrestin1 complexes as well as the studies utilizing biophysical methods, into a structural model. Unfortunately, no structures of such complexes exist for β -arrestin2. Nevertheless, the authors could have capitalized on their finding of the differential conformational changes in these isoforms upon the receptor binding to offer a provisional model for β -arrestin2 in the complex as well. Otherwise, a reader is left with a sense of dissatisfaction of seeing nice diagrams of the sensor responsiveness to the receptor binding but no structural interpretation of these changes and/or differences between the arrestin isoforms.

*We agree with the reviewer's point that a model would help to satisfy the reader's expectations. Hence, we consulted Dr. Xavier Deupi, who is now included in the author list, in order to build a model from existing structures to provide a global structural interpretation of our data in the newly created **Figure 7**. Since the presented β -arrestin1 and 2 conformational change biosensors provide valuable global information about the molecular rearrangements that occur in arrestin proteins in different conditions, we now used this information to visualise our interpretation of the geometry of GPCR- β -arrestin complexes. Additionally, we used our extensive dataset, including recruitment, microscopy and conformational change data to point out differences in the formation of "hanging" and "core" complex configurations between β -arrestin1 and 2. All this information was used to create a model of these differently formed complexes to finally sum up the manuscript as part of the discussion section (lines 542-574). We thank the reviewer for this suggestion and are certain that this addition will aid in the comprehension of our findings.*

- Another issue is the distinction between the "hanging" and "core" complexes. Although it has been reported that arrestins could form such distinct complexes, it seems that the issue has been somewhat overinterpreted in this case. First, deletion of all GRKs essentially abolished the recruitment of both arrestins to the WT receptor, which suggests that at least this receptor does not form "core" complexes independently of phosphorylation. Second, the defects seen in mutants with the deletion of the finger loop (like in β -arrestin2 in Fig 1c) do not necessarily mean that the arrestin recruitment relies on "both GRK-mediated receptor phosphorylation, as well as

the FLR interaction interface” as independent entities. The finger loop has been shown to change conformation upon binding to active phosphorylated GPCR (see, for example, here: J Biol Chem, 2014 Jul 25;289(30):20991-1002; Cell, 2020 Dec 23;183(7):1813-1825). It might be that phosphorylation-induced fit in the finger loop contributes to the high-affinity arrestin binding, and the absence of such contribution might be the reason for the reduced functionality.

We thank the reviewer for her or his perceptive critique regarding the interpretation of β -arrestin1 and 2 recruitment in the absence of GRKs and upon deletion of the respective finger loop regions (FLRs). We agree that some of the recruitment data might have been interpreted in too much detail, specifically in a way that would call attention towards miniscule differences between the two β -arrestin isoforms. Our data indeed show that both β -arrestin isoforms interact best with the PTH1R when all binding interfaces are present and functional (WT receptor, WT β -arrestin and WT HEK cells), proposedly via the formation of phosphorylation- and FLR-dependent “core” complexes. Accordingly, we attenuated the phrasing in the main text that explains our conclusions regarding the BRET recruitment data in **Figure 1**. Moreover, we included other interpretations, as described by the referee (lines 144-171).

While we agree that both, the removal of GRKs from the system and deletion of the respective FLRs, have staggering effects on the recruitment of both β -arrestin isoforms to the PTH1R, we would like to argue that none of these conditions abolishes the recruitment completely. To better show the apparent differences between the PTH1R recruitment behaviours of β -arrestin1 and 2, we presented all three conditions in separate graphs with appropriate y-axis limits. For the reviewer’s convenience we added this representation to the point-to-point reply, as well as in **Supplementary Figure 1a-c**.

As visible in the figure above, all mentioned conditions yield presentable concentration response curves, which we interpret as recruitment. At 3 μ M PTH(1-34), the two WT β -arrestin constructs register 6.06 (β -arrestin1) and 7.35 (β -arrestin2) fold BRET changes over baseline and vehicle measurements. The β -arrestin constructs missing their respective FLRs (-dFLR) still showed a 2.81 (β -arrestin1-dFLR) and 1.42 (β -arrestin2-dFLR) fold change, while the recruitment is most prominently obstructed upon removal of GRK2,3,5 and 6, in Δ Q-GRK (1.39 fold change for β -arrestin1 and 1.77 fold change for β -arrestin2). With these data, we show that even in the conditions that elicit the lowest recruitment (β -arrestin2-dFLR and β -arrestin1 in Δ Q-GRK), we were still able to register a BRET change of approximately 40%, possibly enabled by the highly optimised NanoLuc Halo618 measurement system. This is, in fact, important for the fidelity of the whole study, since extensive experiments were conducted to characterise β -arrestin conformational changes without the influence of GRK phosphorylation, using the empty Δ Q-GRK background (**Figure 5**).

- The statement that β -arrestin2 engages active GPCR independently of phosphorylation better than β -arrestin1 (page 3 line 159) is not supported by the data presented: in cells lacking GRKs the binding of both isoforms is essentially eliminated (Fig 1c). Furthermore, if anything, β -arrestin2 seems to have higher demands for receptor phosphorylation, since it needs both phosphorylation clusters to be present for the full function (Fig. 4a,e). The authors seem to base this statement on the effect of the finger loop deletion, which is more detrimental in β -arrestin2. However, as mentioned above, this could have alternative interpretations.

We again thank the reviewer for pointing out these inconsistencies. As stated above, we toned down the interpretation of the presented BRET recruitment assays, included additional interpretations (as stated by the reviewer above, lines 144-171) and included the full-scale concentration response curves in **Supplementary Figure 1a-c**. We agree with the referee that our original data representation did not allow for drawing the discussed conclusions, yet we hope that our results can be more easily recognised with the changes we made.

- Fig 1 c: The description of the statistical analysis is unclear: If Dunnett's test was used, how is the significance between dFLP and WT in ΔQ was arrived at? Dunnett's test compares each treatment to a single control.

We thank the reviewer for spotting this mistake. The statistical analysis of **Figure 1c** was indeed not performed by using Dunnett's test, but via a two-sided Tukey's test. This error has been corrected in the revised version of the manuscript, as the figure legend now accurately states the utilised statistical test.

- The "hanging" conformation was originally proposed as the conformation competent to support receptor internalization and arrestin signaling but not receptor desensitization (Proc Natl Acad Sci U S A. 2017 Mar 7; 114(10): 2562–2567). This does not seem to be the case here, as barr2-dFLP essentially does not support receptor internalization, and the results are unclear for barr1-dFLP: Fig 1f: The difference between barr1-dFLR and no barr1 is not shown as statistically significant. How would the authors interpret that in view of their data?

The publication of Cahill III et al. 2017 exclusively focused on β -arrestin1 and to our knowledge, the dFLR mutants for both β -arrestin isoforms have not yet been characterised side-by-side, regarding their functionality. Hence, we would like to argue that our results are in line with the mentioned publication, as we show that β -arrestin1-dFLR construct is able to increase PTH1R internalisation in comparison to the drastically reduced internalisation in ΔQ -GRK. As the reviewer aptly states, the numerical and visual difference in PTH1R co-localisation with Rab5 in the absence of β -arrestins compared to the re-expression of β -arrestin1-dFLR (**Figure 1f, i**) is not significant (**Supplementary Table 1**). From this, we conclude that the β -arrestin1-dFLR mutant is not fully capable of rescuing the phenotype of the β -arrestin1/2 knockout cell line, but most importantly, these experiments show that β -arrestins and GRKs have to synergistically act on activated GPCRs in order to mediate receptor internalisation. Specifically, the increase in PTH1R co-localisation with Rab5 measured in the absence of β -arrestins (as compared to the data that was recorded in ΔQ -GRK) shows that GRKs play an important role in all internalisation and trafficking events, while our measurements in **Figure 6** indicate that β -arrestins are as crucial as GRKs for endosomal delivery of the PTH1R.

- Fig. 1f,i: There is no significant difference between no-barr and ΔQ -GRK for either b-arrestin1 or b-arrestin2. Therefore, the statement that "other phosphorylation-dependent but b-arrestin-independent ways must exist to internalize the PTH1R" is premature. This statement is contradicted by the authors' own data showing that the removal of either b-arrestins or GRKs has equally detrimental effect of the PTH1R trafficking to endosomes (Fig. 6d).

We thank the reviewer for this insightful comment. We agree that the differentiation between phosphorylation-dependent and β -arrestin-dependent events is problematic as these processes happen concomitantly and necessitate each other to a certain degree. In line with the argumentation above, we changed the statement to "These measurements imply that β -arrestins and GRKs have to act synergistically on activated GPCRs to mediate receptor internalisation." (lines 207-216).

- Figs 1d,g: There does not seem to be quantification for these data, which makes it impossible to compare arrestin recruitment/internalization behavior with the receptor internalization data. Furthermore - and this applies to Figs 1d,g,f, I and Suppl Table 1 - it is unclear how the cells have been selected for the analysis from how many slides, experiments, etc. Additionally, these

experiments need the data on the expression levels of all proteins involved, because that would heavily influence the image analysis data, particularly because different modified cell lines are involved.

*In **Figure 1**, we abstained from showing β -arrestin-YFP co-localisation data, as we already provide a β -arrestin recruitment dataset using NanoBRET (**Figure 1b** and **c**). Co-localisation between β -arrestin-YFP and the PTH1R-CFP is shown in **Supplementary Figure 1f** and **g**. Here, we would like to thank the reviewer for expressing her or his concern, as we added the missing β -arrestin-YFP-PTH1R-CFP co-localisation in Δ Q-GRK upon addressing this comment.*

*In general, for our live-cell confocal microscopy experiments, we acquired between 5 and 10 images per condition from at least three cover slips, prepared from at least three independent transfections ($n \geq 3$ with a total image number of ≥ 15). There was no specific cell selection process, other than all three fluorescent proteins had to be visible under the unchanged laser intensity and gain settings. To address the referee's comment, we included the sentence "Images were acquired before and after stimulation with 100 nM PTH(1-34) for 15 minutes from at least three cover slips, prepared from at least three independent transfections ($n \geq 3$)." to the legend of **Figure 1d** and **g**. Additionally, the exact number of paired images used for all quantifications of co-localisation is stated in the legend of **Figure 1f** and **i**: (number of images per respective condition; β arr1 (39), β arr1-dFLR (43), no β arr (17), β arr1 in Δ Q-GRK (38), β arr2 (33), β arr2-dFLR (27), β arr2 in Δ Q-GRK (50)).*

*Furthermore, we re-analysed all images and extracted the characteristic mean object intensity and total object signal per cell size recorded for PTH1R-CFP, β -arrestin-YFP and Rab5-mCherry. These data should enable the at-a-glance evaluation of sample size, as well as transfection, image and segmentation quality (basal images only). In the revised **Supplementary Figure 1h** and **i** (which we did not include in the point-to-point reply due to its size), these values are plotted for each individual basal image used for quantification. Specifically, the PTH1R and Rab5 object intensity does not seem to vary between the different conditions. In our analysis, we found that β -arrestin1-dFLR-YFP shows a slightly higher mean object intensity than β -arrestin1-YFP. Still, we would like to argue that neither this change, which possibly results from a slightly different expression behaviour of the mutant protein, nor any other shown signal spread would be suitable to explain the measured differences in PTH1R-Rab5 co-localisation shown in **Figure 1f** and **i** (lines 216-220). We hope that this in-depth analysis provides additional insight into our microscopy method and alleviates the reviewer's concerns.*

- Suppl Fig 7: GRK expression levels? How the levels compare among different GRK isoforms and between the experiments with b-arrestin1 and b-arrestin2? It seems that no individual GRK is capable of restoring the recruitment of b-arrestin2 to PTR1R-WT, whereas any is sufficient to restore the recruitment of b-arrestin1. However, this could be simply due to inconsistencies in the expression levels.

*We thank the reviewer for this perceptive comment and agree with the assessment that already slight changes in GRK expression could be responsible for the observed differences between β -arrestin1 and 2, regarding their GRK-specific recruitment to the PTH1R. The basis for our GRK-specific β -arrestin recruitment assay utilising Δ Q-GRK was just recently published with Nature Communications¹. In this publication we quantified the GRK overexpression in order to validate the data generated for 12 different GPCRs and we were not able to detect any systematic errors between the GRK-specific recruitment of β -arrestin1 and 2. In **Supplementary Figure 3** of Drube et al.¹ we were able to characterise the catalytic function of YFP-tagged GRKs to be similar to their untagged counterparts. For the reviewer's convenience, we included this panel in the point-to-point reply:*

Supplementary Figure 3

Drube et al.¹, 2022

To directly address the referee's question, we performed an additional experiment and quantified the fluorescence of YFP-tagged GRK2, 3, 5 and 6 using analogous transfection schemes as in our GRK-specific β -arrestin1 and 2 recruitment assay. The absolute YFP fluorescence and relative values, normalised to the respective β -arrestin1 condition, derived from measurements of 32 technical replicates per condition, can now be accessed in **Supplementary Figure 8** and the point-to-point reply below:

Together with the validation experiments performed in our previous publication, we argue that the data in **Supplementary Figure 8** show that the degree of overexpression is similar for all four tested GRK isoforms and between the β -arrestin1 and 2 conditions. Conclusively, we cited the preceding publication and referenced our newly generated data at the appropriate position in the manuscript (lines 327-330 and lines 335-336).

- Fig 6g: Arrestin-dependent ERK activation is diminished when induced via PTH1R-PD2 as compared to the WT receptor. The authors suggest that the reduced capacity of PTH1R-PD2 to localize to endosomes might be directly responsible for this defect. Although arrestin could drive the ERK activation from endosomes, it is probably not always the case. There is strong evidence that only receptor-bound arrestins activate ERK. Thus, a simpler alternative explanation would be that PTH1R-PD2 is less efficient in activating ERK via arrestins because it is less efficient in recruiting arrestins.

We thank the referee for this intelligible comment. The reviewer indeed raised an important point by stating that the reduced β -arrestin recruitment profile of the PTH1R-PD2 receptor variant could serve as the most straightforward explanation for its attenuated ERK1/2 signalling behaviour. Hence, we included this interpretation as the leading phrase discussing the impact of our phospho-ERK1/2 analysis in the main text (lines 508-512).

Reviewer #2 (Remarks to the Author):

The manuscript by Haider et al. focuses on the two isoforms of β -arrestins (1 and 2), their ability to elicit distinct conformations following their interaction with agonist-stimulated GPCR, in this case, the PTH1R in living cells. In particular, the authors investigated the interplay between specific β -arrestin1/2 conformations and different C-terminus receptor phosphorylation patterns (full, partial at proximal, partial at distal, or altogether none via CRISPR Δ Q-GRK cells). This study extends Cahill III et al.'s work from 2017 (focused on β -arrestin1 isoform only) by investigating the two activation pathways and agonist-activated C-terminus phosphorylation pattern dependence corresponding to 'hanging', and 'core' GPCR- β -arrestin conformations utilizing the two β -arrestin isoforms (1 and 2). The authors employed a combination of BRET-based β -arrestin recruitment, internalization, β -arrestin conformational changes (NanoLuc/FIAsH)-biosensors, confocal microscopy, and signaling downstream of PTH1R (ERK1/2). The results from this study raise exciting opportunities for exploring the phosphorylation barcode for different signaling outcomes 'based signaling'; and the conformational and signaling outputs by the two isoforms of β -arrestin. Overall, the article is well written, the experiments are well designed and rigorously analyzed, and the data broadly support the conclusions. I believe it will be of broad interest to scientists studying GPCR signaling, biased agonism, and GPCR pharmacology. I recommend its publication in Nature Communications after the authors clarify the questions raised below:

We thank the reviewer for her or his positive assessment of our study. Encouraged by the referee's comprehensive summary of our work and findings, we hope that the following point-to-point reply satisfyingly resolves the fair questions raised.

1. The authors observe distinct conformational signatures on β -arrestins dictated by differentially phosphorylated receptors using the BRET-based conformational biosensors. As well as how these differentially phosphorylated receptor patterns engage with WT β -arrestin 1/2 or their finger loop deleted counterparts to deduce the "core" and the "hanging" complexes. While all this is fascinating, it is still not clear (for example, in lines 145-156 and elsewhere throughout the manuscript) what the contribution of each β -arrestin isoform (say percentage) is in inducing particularly the "core" or "hanging" (as in Δ Q-GRK cells) conformation? I feel quantification of these specific conformations is essential in this study. For example, in line 146, the authors indicate they obtained ~36% of "hanging" complexes when using β -arrestin1 isoform. Throughout the manuscript, the contribution of a specific isoform to these two conformations, particularly on the 'core' confirmation, wasn't clear. Overall it appears to be minor (or perhaps very difficult to deduce) from these BRET-based data? The authors need to clarify this throughout the manuscript. What would the β -arrestin 1/2 conformational signatures pattern in Fig. 3 be with a different receptor (another class B or just class A)?

*We thank the referee for her or his keen observations and the comments about how we interpret the recruitment of β -arrestins in different conditions (WT, -dFLR or in Δ Q-GRK), in order to characterise the configuration of formed PTH1R- β -arrestin complexes. As the reviewer already expressed, with the currently used methods it is difficult to infer the exact percentage of complexes that form in "core" and "hanging" configurations. Moreover, our BRET measurements aimed to assess intrinsic characteristics of the two β -arrestin isoforms. Resulting from the data presented in **Figure 1b** and **c**, we expect the majority of PTH1R- β -arrestin complexes to form in a tight configuration that involves both, β -arrestin association with the phosphorylated PTH1R C-terminus and the insertion of the FLR into the active GPCR cavity. This conclusion can be drawn as the deletion of the FLR or removal of GRKs drastically reduces the recruitment for both isoforms. However, all tested conditions still yield quantifiable β -arrestin recruitment and we found detailed differences between β -arrestin1 and 2 and their capabilities to form either "hanging" or phosphorylation-independent "core" complexes. Here, β -arrestin1-dFLR (produces still 36% of the β -arrestin1 recruitment) seems to be better suited to bind the PTH1R in a presumably*

“hanging” complex, in comparison to β -arrestin2-dFLR, which only registered around 7% of the measured β -arrestin2 recruitment. This behaviour seems to be reversed in Δ Q-GRK, leading us to the conclusion that β -arrestin2 is more readily activated by active, yet unphosphorylated GPCRs, in comparison to β -arrestin1. These findings shed light on the molecular capabilities of β -arrestin1 and 2 to form complexes with GPCRs in different configurations, while we still assume that the analysed PTH1R- β -arrestin complexes employ all available binding interfaces (hence, phosphorylation-dependent “core” complexes, as long as GRKs are present in the system).

To address the intelligible concerns raised by the reviewer, we particularly changed the main text to better reflect the ambiguity of our findings regarding the absolute percentage of differentially formed complexes. Additionally, we attenuated our interpretation of these results (lines 144-171) and added the full-scale recruitment curves (already present in **Figure 1b**) to **Supplementary Figure 1a-c**. Moreover, initiated by the comments of reviewer 1, we now also included a structural modelling of “hanging” and “core” complexes, which are differently formed for β -arrestin1 and 2 in the novel **Figure 7**. We sincerely believe that these changes will be very helpful for the reader and thank the referee for her or his suggestions.

To answer the last question, whether β -arrestin conformational change signatures would differ for the binding to different GPCRs, we would kindly divert the reviewer’s attention towards the publication of the original sensor design^{2,3}. These studies utilise analogue measuring systems (FRET and BRET) and show that β -arrestin2 conformational change signatures are, in fact, specific for each tested GPCR. Although, this has not been shown for β -arrestin1 via the use of adequate biosensors, we strongly believe that this behaviour is shared between the two isoforms. In conclusion, the study at hand focusses on differences in conformational change between the two isoforms for binding to the same GPCR. Furthermore, we would like to argue that providing β -arrestin1 and 2 conformational change data for another (possibly class A) GPCR might reveal new and interesting biological features but would not add to the main conclusions of this study per se.

2. The conclusion on signaling competence of receptor- β -arrestin conformations on downstream ERK1/2 phosphorylation is only done in the context of the receptor's phosphorylation patterns (WT, PD1, and PD2). The study did not include what may be happening in ERK1/2 signaling in the context of two isoforms of β -arrestin vis-a-vis the phosphorylation patterns. Such data would strengthen the manuscript significantly.

We thank the reviewer for this insightful suggestion. Indeed, we agree that more research needs to be done to finally delineate the roles of individual β -arrestin isoforms regarding the proposed amplification of ERK1/2 phosphorylation. Following the referee’s recommendation we attempted to rescue the apparent effect of β -arrestin 1 and 2 knockout on the ERK signalling capacity of the PTH1R, as shown in **Figure 6g** and **h**. For this, we reintroduced each β -arrestin isoform individually in β -arrestin1/2 KO cells and expected an increase in the cellular pERK1/2 response upon agonist activation, in comparison to the β -arrestin devoid condition. We included a representative blot, as well as the quantification of three independent experiments below.

As it is apparent in the representative blot and in the quantification, our efforts to increase the pERK1/2 response via overexpression of either β -arrestin 1 or 2 yielded virtually indistinguishable signals, in comparison to the β -arrestin1/2 KO + empty vector condition. This result was quite surprising to us, yet after reassessment of the related literature, we still want to provide several possible explanations for this phenomenon.

We performed all of our pERK1/2 analyses using transient PEI transfection. The experiments shown in **Figure 6g** and **h** only rely on the overexpression of the probed receptor variant. Even though our transfection methods usually yield a homogenous pattern of protein overexpression (**Supplementary Figure 1h, i** and **Supplementary Figure 8**), we cannot control whether the additionally co-expressed β -arrestin constructs are present in all individual cells that respond upon agonist addition, especially in ensemble Western blot measurements. Since β -arrestins are hypothesised to play an auxiliary role⁴ in the amplification of GPCR-induced pERK1/2 signalling, the possibly heterogeneous expression of these proteins might be enough to occlude most of the expected signal increase. This complication would argue for the establishment and utilisation of expression level-controlled stable cells or single β -arrestin isoform KO cell lines, which are not readily available in our laboratory.

Furthermore, it might also be possible that both β -arrestin isoforms are needed for efficient pERK1/2 signal amplification, or that we are observing even more complex processes. Multiple publications address this and discuss the functionality of β -arrestins in regards to ERK signalling^{3,5}. The latter study even compared β -arrestin knockdown approaches and β -arrestin KO cell lines and similarly attempted rescue experiments by overexpression of individual or even both β -arrestin isoforms, with mixed success. Following their analysis, it seems that specifically the rescue experiments yielded different results depending on the utilised cell line. In contrast to the PTH1R data presented in our **Figure 6g** and **h**, the authors show that the complete knockout of β -arrestin1 and 2 increased the ERK1/2 signalling capacity of endogenous β 2 adrenergic receptor (their Figure 1a), yet they were similarly not able to reverse this effect upon β -arrestin1/2 overexpression (their Figure 3d), utilising the identical knockout cell line as we do (Asuka Inoue). These experiments seem to be complicated due to the proposed dual functionality of β -arrestins. On the one hand, arrestins dampen G protein activation via desensitisation of active and phosphorylated GPCRs^{6,7} (hence, decreasing the ERK1/2 activation capacity of a given receptor), while on the other hand they might play an important role as scaffolds that bring the active components of the MAPK signalling pathway in close proximity of each other⁸ (hence, amplifying the turnover of mutually activating kinases to increase ERK1/2 phosphorylation). Depending on the degree of overexpression, one of these functions might be overemphasised, or these effects might ultimately cancel each other out, obstructing a clear assessment using the gold-standard Western blot method.

We would like to argue that a detailed reassessment of these pivotal signalling events is needed to finally unravel the interplay between the activation of β -arrestins and the concomitant propagation of MAPK signalling. In this respect, we are working on the establishment of suitable cellular systems (e.g. GRK + (single) β -arrestin KO cell lines) and the acquisition of compartmentalised pERK1/2 biosensors⁹ to investigate the spatiotemporal signature of GPCR-mediated ERK1/2 signalling. Yet, we believe that

the definite clarification of these highly relevant and interesting biological phenomena requires multiple orthogonal measuring approaches, which would exceed the scope of the manuscript at hand.

3. Minor: Overall, the figures look great, but some of the labels' font size and color choices (e.g., yellow) are hard to visualize. The authors may need re-formatting the figures and enlarge the labels' relative sizes. I had so much difficulty reading the labels, for example, in Figures: 2, 4, and 6 and in many SI ones. For instance, what EV in the figures, as in 4 implies, wasn't described in the figure legends.

We thank the referee for expressing her or his appreciation of our visual data representation. Furthermore we agree with the referee's evaluation regarding issues with figure label sizes and shortcomings in the figure legends. To address these points, we raised the font size of labels to a minimum of 5 pt. Furthermore, all figure legends were checked again and adjusted to include missing abbreviations. A full list of changes to display items can be accessed at the beginning of this point-to-point reply. We hope that with these changes we improved the intelligibility and accessibility of our work and successfully addressed the reviewer's concerns.

References

- 1 Drube, J. *et al.* GPCR kinase knockout cells reveal the impact of individual GRKs on arrestin binding and GPCR regulation. *Nat Commun* **13**, 540, doi:10.1038/s41467-022-28152-8 (2022).
- 2 Nuber, S. *et al.* beta-Arrestin biosensors reveal a rapid, receptor-dependent activation/deactivation cycle. *Nature* **531**, 661-664, doi:10.1038/nature17198 (2016).
- 3 Lee, M. H. *et al.* The conformational signature of beta-arrestin2 predicts its trafficking and signalling functions. *Nature* **531**, 665-668, doi:10.1038/nature17154 (2016).
- 4 Grundmann, M. *et al.* Lack of beta-arrestin signaling in the absence of active G proteins. *Nat Commun* **9**, 341, doi:10.1038/s41467-017-02661-3 (2018).
- 5 Luttrell, L. M. *et al.* Manifold roles of beta-arrestins in GPCR signaling elucidated with siRNA and CRISPR/Cas9. *Sci Signal* **11**, doi:10.1126/scisignal.aat7650 (2018).
- 6 Ferguson, S. S. Evolving concepts in G protein-coupled receptor endocytosis: the role in receptor desensitization and signaling. *Pharmacol Rev* **53**, 1-24 (2001).
- 7 Peterson, Y. K. & Luttrell, L. M. The Diverse Roles of Arrestin Scaffolds in G Protein-Coupled Receptor Signaling. *Pharmacol Rev* **69**, 256-297, doi:10.1124/pr.116.013367 (2017).
- 8 Bourquard, T. *et al.* Unraveling the molecular architecture of a G protein-coupled receptor/beta-arrestin/Erk module complex. *Sci Rep* **5**, 10760, doi:10.1038/srep10760 (2015).
- 9 Keyes, J. *et al.* Signaling diversity enabled by Rap1-regulated plasma membrane ERK with distinct temporal dynamics. *Elife* **9**, doi:10.7554/eLife.57410 (2020).

Reviewer #1 (Remarks to the Author):

All my critics have been answered, and I have no additional concerns. The authors have made substantial revisions to the manuscript, which have significantly improved the paper. It is an excellent work and should be of great interest to the field.

Reviewer #2 (Remarks to the Author):

The authors have addressed all my comments and concerns in the revised manuscript. I have no further comments.